# Revealing the role of interfacial water and key intermediates at ruthenium surfaces in the alkaline hydrogen evolution reaction

Xing Chen[1,5], Xiao-Ting Wang[1,5], Jia-Bo Le[2,5], Shu-Min Li[1], Xue Wang[2], Yu-Jin Zhang[1], Petar Radjenovic[1], Yu Zhao[1], Yao-Hui Wang[1], Xiu-Mei Lin [1,3] ✉, Jin-Chao Dong [1,4] ✉ & Jian-Feng Li [1,4] ✉

Ruthenium exhibits comparable or even better alkaline hydrogen evolution reaction activity than platinum, however, the mechanistic aspects are yet to be settled, which are elucidated by combining in situ Raman spectroscopy and theoretical calculations herein. We simultaneously capture dynamic spectral evidence of Ru surfaces, interfacial water, *H and *OH intermediates. Ru surfaces exist in different valence states in the reaction potential range, dissociating interfacial water differently and generating two distinct *H, resulting in different activities. The local cation tuning effect of hydrated Na⁺ ion water and the large work function of high-valence Ru(n+) surfaces promote interfacial water dissociation. Moreover, compared to low-valence Ru(0) surfaces, high-valence Ru(n+) surfaces have more moderate adsorption energies for interfacial water, *H, and *OH. They, therefore, facilitate the activity. Our findings demonstrate the regulation of valence state on interfacial water, intermediates, and finally the catalytic activity, which provide guidelines for the rational design of high-efficiency catalysts.

The detailed structure of interfacial water and key intermediates significantly influences the electrochemical performances. A clear understanding of the chemical state (composition and valence)−activity relationships and reaction mechanisms of the hydrogen evolution reaction (HER) are vital for designing high-efficiency catalysts to realize a hydrogen economy towards sustainable energy development[1–5]. The HER pathway in both acidic and alkaline electrolytes involves Volmer-Heyrovsky or Volmer-Tafel steps, except that in the former the adsorbed hydrogen (*H) is from hydronium ions ($H_3O^+$)[6–12] while in the latter the *H are formed by an initial water dissociation step ($H_2O + e^- \rightarrow *H + OH^-$). This water dissociation step may introduce an additional energy barrier, resulting in sluggish kinetics[12–16]. Ru with only 1/4 price of

Pt exhibits comparable or even better alkaline HER activity than Pt[17–20], however, the mechanistic aspects are yet to be settled. Significantly, the activity of a catalyst is strongly related to its valence state and it is known that Ru is easily oxidized to ruthenium oxides with different valence states ($RuO_x$). It is reported that the high oxygen affinity and the low energy barrier of Ru can support $H_2O$ dissociation, and the H-adsorption and/or OH-adsorption energies then further influence the alkaline HER activity, however, by far no direct spectral evidence of Ru valence−interfacial water and intermediates−HER activity relationships have been directly observed[21–23].

Traditional electrochemical techniques afford deep insights into the kinetics of catalytic reactions while molecular information about

[1]College of Energy, College of Chemistry and Chemical Engineering, College of Materials, State Key Laboratory of Physical Chemistry of Solid Surfaces, iChEM, Xiamen University, Xiamen 361005, China. [2]Key Laboratory of Advanced Fuel Cells and Electrolyzers Technology of Zhejiang Province, Ningbo Institute of Materials Technology and Engineering, Chinese Academy of Sciences, Ningbo 315201, China. [3]Department of Chemistry and Environment Science, Fujian Province University Key Laboratory of Analytical Science, Minnan Normal University, Zhangzhou 363000, China. [4]Innovation Laboratory for Sciences and Technologies of Energy Materials of Fujian Province (IKKEM), Xiamen 361005, China. [5]These authors contributed equally: Xing Chen, Xiao-Ting Wang, Jia-Bo Le. ✉e-mail: xiu-mei.lin@xmu.edu.cn; jcdong@xmu.edu.cn; Li@xmu.edu.cn

the species is lacking[24–26]. Spectroscopic techniques can provide molecular information about the species[27–29]. However, due to technical limitations, it is challenging to simultaneously acquire molecular fingerprint information of catalyst surfaces, interfacial water, and *H, and *OH/OH⁻ intermediates in aqueous solutions during alkaline HER, using conventional experimental spectroscopic techniques, such as Raman spectroscopy, infrared spectroscopy, sum frequency generation spectroscopy, and X-ray spectroscopy.

Core-shell nanoparticle-enhanced Raman spectroscopy can provide spectral information over the wavenumber range from ~100 cm⁻¹ to 4000 cm⁻¹ at a single molecule sensitivity due to the plasmonic enhancement effect of the Au core. It allows the acquisition of the spectral signals in the low wavenumber region (<800 cm⁻¹), where most oxygen species, hydroxyl groups, and metal-oxide bonds occur[30–32]. It was therefore employed to in situ track the alkaline HER process at Ru catalyst surfaces by constructing Au@Ru core-shell nanoparticles (NPs) in this work. We simultaneously captured dynamic spectral evidence of Ru catalyst surfaces in different valence states ($RuO_x$), three different structures of interfacial water (4-HB·$H_2O$, 2-HB·$H_2O$, and Na·$H_2O$), two distinct *H and *OH intermediates, and the interactions between them. Their distinct roles in alkaline HER were then revealed, allowing for valence–interfacial water and intermediates–activity relationships and alkaline HER mechanism understanding with catalytic activity improvement.

## Results and discussion

### Characterization of Au@Ru core-shell nanoparticles (NPs)

Core-shell nanoparticle-enhanced Raman spectroscopy was employed to investigate the alkaline HER process by constructing Au@Ru core-shell NPs illustrated in Fig. 1a. Supplementary Fig. 1a, b shows the SEM and TEM images of the synthesized Au@Ru NPs, respectively. The NPs possessed a uniform size and good dispersibility with a diameter of the Au core of ~55 nm and a thickness of the Ru shell of ~2.5 nm, and no obvious pinholes can be observed. HAADF-STEM and elemental mapping images show the uniform distribution of Au and Ru in the NPs (Fig. 1b–e). In Supplementary Fig. 2, a comparison of cyclic

voltammogram curves of 55 nm Au NPs (a, red) and 55 nm Au@2.5 nm Ru NPs (a, blue) and core-shell nanoparticle-enhanced Raman spectroscopy of CO adsorbed on 55 nm Au@0.7 nm Ru NPs (b, red) and 55 nm Au@2.5 nm Ru NPs (b, black) demonstrate the pinhole-free structure of the 55 nm Au@2.5 nm Ru NPs. Supplementary Fig. 3a–c shows the exponential decrease of the Raman intensity with increasing the Ru shell thickness. The 3D-FDTD calculations prove that the corresponding highest enhancement of the Raman signal is $7 \times 10^6$ for the 55 nm Au@2.5 nm Ru NPs. The strong enhancement effect of the electric field significantly amplifies the Raman signal of the species in our systems. They are therefore detected. In Supplementary Fig. 4, XPS spectra of the Ru *3p* and O *1s* electrons of the Au@Ru nanocatalysts show the oxidation of the most out-layer of the Ru shell under atmospheric conditions. Therefore, different Ru valences are expected to present on the Au@Ru NPs. Detailed analysis of Supplementary Figs. 2–4 please see supplementary information. In brief, the physical-chemical characterizations and theoretical calculations demonstrate the compact and pinhole-free structure of the 55 nm Au@2.5 nm Ru core-shell NPs for effective HER catalysis and enhanced Raman signal capturing in our systems.

### HER process at Ru surfaces under alkaline conditions

First, we tested the HER performance of 55 nm Au and 55 nm Au@2.5 nm Ru NPs in a 0.1 M NaOH solution (pH ~13). As shown in Fig. 2a, the Au@Ru NPs have an overpotential of -176 mV at a current density of −10 mA cm⁻² while Au NPs have far inferior HER performance under the same conditions. Here, in all the polarization curves of the HER process, the current density is calculated based on the geometric area of the electrode. Thus, a remarkably improved HER performance is achieved with the 2.5 nm Ru shell coating on the Au cores. We also tested the HER performance of Au@Ru NPs in a typical 1.0 M KOH solution and 1.0 M NaOH solution (Supplementary Fig. 5). The overpotentials of alkaline HER in 1.0 M KOH and NaOH are similar (67 mV), which is comparable to the reported results[33] and demonstrates the generality of the Au@Ru nanocatalysts. But they are much smaller than in 0.1 M NaOH (176 mV), indicating the faster reaction

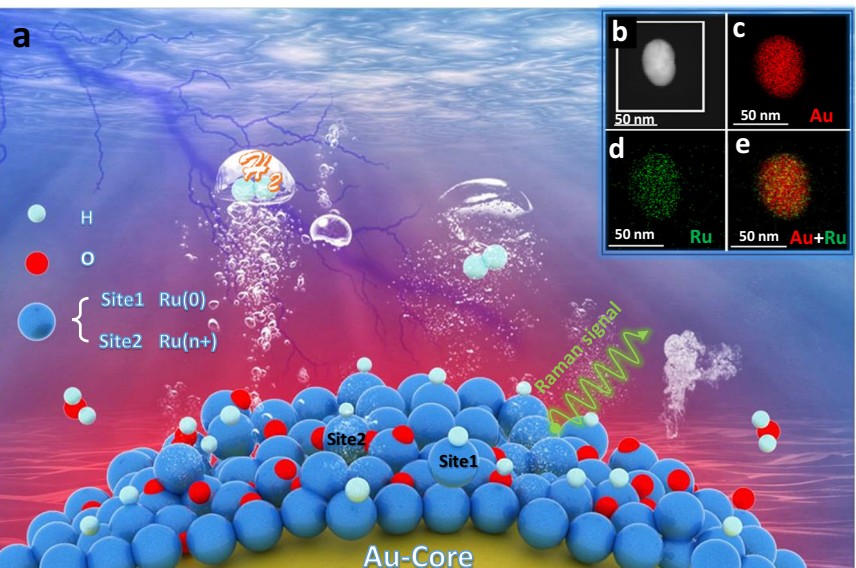

**Fig. 1 | Schematic illustration of the core-shell nanoparticle-enhanced Raman spectroscopy study of the alkaline HER process and correlated characterization of 55 nm Au@2.5 nm Ru core-shell NPs. a** Model of 55 nm Au@2.5 nm Ru core-shell NPs and the mechanism of the alkaline HER process revealed by core-shell nanoparticle-enhanced Raman spectroscopy. The golden, blue, red, and light-blue spheres represent Au, Ru, O, and H atoms, respectively. The contiguous pinhole-free 2.5 nm Ru shell protects the Au core from contact with the environment and provides Ru active sites for the catalysis of HER. The Au@Ru core-shell NPs, when being illuminated by an incident laser, the Au core can generate strong electromagnetic fields to enhance the Raman signals of molecules adsorbed at the Ru shell catalyst surfaces. **b–e** Electron microscope characterization of 55 nm Au@2.5 nm Ru NPs, insets: **b** High-angle annular dark field scanning transmission electron microscopy (HAADF-STEM) image of a 55 nm Au@2.5 nm Ru NP. **c–e** Au, Ru, and Au + Ru elemental mapping images of a 55 nm Au@2.5 nm Ru NP.

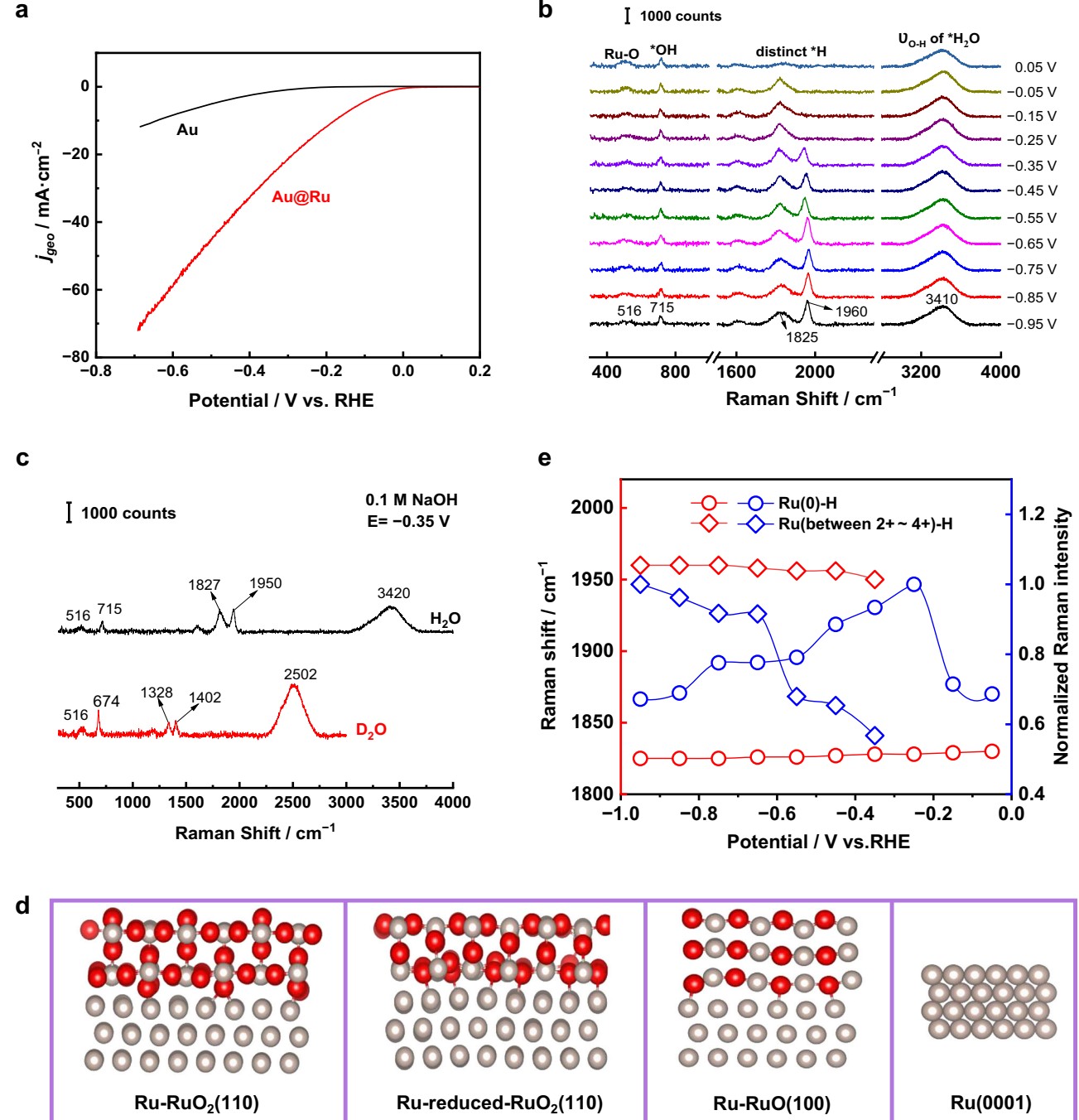

**Fig. 2 | Tracking alkaline HER process at Ru surfaces by in situ Raman spectroscopy and the correlated DFT results. a** Polarization curves of the HER process at 55 nm Au and 55 nm Au@2.5 nm Ru surfaces in 0.1 M NaOH (Ar-saturated) with iR correction (i, current; R, resistance), 5 mV/s scanning rate and 1600 r.p.m. **b** In situ Raman spectra of the alkaline HER process at 55 nm Au@2.5 nm Ru electrode surfaces. **c** Raman spectrum (black) and Deuterium isotopic substitution Raman spectrum (red) of the alkaline HER at 55 nm Au@2.5 nm Ru surfaces at −0.35 V. **d** Models of Ru-RuO$_2$(110), Ru-reduced-RuO$_2$(110), Ru-RuO(100), and Ru(0001) surfaces. Ru: gray spheres; O: red spheres. **e** Normalized Raman intensities (blue) and frequency shifts (red) of the Ru-H band at low-valence state Ru(0) (circle) and high-valence state Ru (between +2 ~ +4) (square) surfaces in the HER potential range. The solution is 0.1 M NaOH saturated with Ar.

kinetics in 1.0 M electrolytes. Therefore, lots of bubbles were generated in a short time due to the fast HER rate under the 1.0 M KOH condition, especially at extremely negative potentials, which influenced the stability of the system for spectra acquisition.

We therefore performed in situ electrochemical core-shell nanoparticle-enhanced Raman measurements on 55 nm Au@2.5 nm Ru NPs in 0.1 M NaOH to track the HER process at Ru surfaces for mechanism exploration. In addition, the Raman spectra were recorded from negative to positive potential to avoid the interference of the

generated H$_2$ gas on the measurements as well. Raman bands around 516, 715, 1825, 1960, and 3410 cm$^{-1}$ are observed at the initial HER potential of −0.95 V vs. RHE. By shifting the potential positively from −0.95 V to −0.35 V, the vibrational frequencies for the bands around 516 cm$^{-1}$ and 715 cm$^{-1}$ are relatively consistent while that for the bands around 1825, 1960, and 3410 cm$^{-1}$ shift to around 1827, 1950, and 3420 cm$^{-1}$ at the potential of −0.35 V (Fig. 2b).

A deuterium isotopic experiment was performed in the complete HER potential range (Supplementary Fig. 6) and potential at −0.35 V

was selected to verify the species observed in Fig. 2b. Substituting $D_2O$ for $H_2O$, the Raman band around 516 cm$^{-1}$ remains stable, indicating that this species is not related to a "H" atom and could be attributed to RuO$_x$ surface oxide originating from the Ru catalysts. Nevertheless, the Raman bands around 715, 1827, 1950, and 3420 cm$^{-1}$ shift to around 674, 1328, 1402, and 2502 cm$^{-1}$, respectively, which imply that these species include the "H" atom. The shifts of these vibrational frequencies accord with the expected shifts from the mass conversion of the formula ($\gamma = 71.1\%$) (as calculated, see mass formula section of Supplementary Information) and previous reports in the literature[27,34,35].

The Raman bands around 1827 and 3420 cm$^{-1}$ can be attributed to the vibrations of the Ru-H bond of adsorbed hydrogen (*H) and interfacial water at the metallic Ru(0) surfaces, respectively[27,34]. To the best of our knowledge, the Raman band around 1950 cm$^{-1}$ at Ru surfaces during HER has not been reported, which was preliminarily attributed to the Ru-H bond of adsorbed hydrogen (*H) at oxidized Ru(n+) surfaces. The oxidation of the most out layer of the Ru shell has been demonstrated by XPS spectra in Supplementary Fig. 4. DFT calculations were carried out (Supplementary Table 1, Fig. 2d, and Supplementary Fig. 7) to further confirm the attribution of the correlated Raman bands and it was found that the frequency of the Ru-H stretching mode increases with an increase in the valence state of Ru, which is consistent with previous publication[36] that the force constant of Ru-H bond is larger for Ru(n+) with higher electronegativity. The calculated vibrational frequencies of the *H metal-adsorption bonds on Ru(4+), Ru(close to 4+), and Ru(2+) active sites are around 1987, 1991, and 1947 cm$^{-1}$, respectively, which match well with our experimental results. Therefore, the band around 1960 cm$^{-1}$ in our experiment can be assigned to metal-adsorption bond vibration of *H at RuO$_x$ surfaces. Furthermore, as shown in Supplementary Fig. 6, during the HER potential range from −0.95 V to 0.05 V, generally, the changing trend of the Raman spectra (appearing or disappearing of the bands, position and intensity of the existing bands) in the $D_2O$ experiment is similar to that in $H_2O$. Our isotope experiment further verifies the phenomenon observed in the $H_2O$ experiment.

Accordingly, Raman bands of two kinds of distinct *H (1825 and 1960 cm$^{-1}$) adsorbed on different valence states Ru surfaces Ru(0/n+) were observed in our Raman spectra. The normalized Raman intensities (blue) and frequency shifts (red) of the Ru-H bond on low-valence state Ru(0) (circle) and high-valence state Ru (between +2 ~ +4) (square) surfaces vs. potentials during the HER are shown in Fig. 2e. With potential increasing, the vibrational frequency of the Ru(n+)-H Raman band (red square) around 1960 cm$^{-1}$ red-shifts and its intensity (blue square) gradually decreases, which could be due to the gradual reduction of the high valence Ru(n+) into zero-valent Ru(0) under the reduction potential. When the control potential is −0.45 V, it is found that the content of high-valence Ru-H gradually decreases with the change of time (Supplementary Fig. 8), which is consistent with the gradual reduction of high-valence Ru at the hydrogen evolution potential. It is worth mentioning here that in the initial state, the catalyst surface contains different valence states of Ru. In the HER potential range, the hypervalent Ru is continuously reduced, and its surface behavior is affected by both potential-induced variations and time-course changes. The vibrational frequency of the Ru(0)-H Raman band (red circle) around 1825 cm$^{-1}$ blue-shifts and its intensity (blue circle) increases first and then decreases, which may be related to the increase of Ru(0) content due to the reduction of high valence Ru(n+). For zero-valent Ru, the surface behavior is mainly affected by potential illustrated by the Stark shift of Ru(0)-H. The normalized Raman intensity and frequency shifts of the Ru-O band versus the HER potentials have been plotted and shown in Supplementary Fig. 9. In the potential range of −0.95 V to −0.35 V, the intensity of the Ru-O band decreases whereas its position keeps constant, which is consistent with the changing of the Ru(n+)-H. It further indicates that the higher

wavenumber Ru-H Raman band is related to the higher valence state Ru. As the potential shifts positively, the degree of Ru-O reduction decreases, leading to the increase of the Ru-O content till the ending potential of 0.05 V.

To further experimentally verify that the bands around 1825 cm$^{-1}$ and 1960 cm$^{-1}$ (Fig. 2b) are two distinct *H species at Ru surface active sites in different valence states, in situ Raman measurements were performed after reduction (Fig. 3a) and oxidation (Fig. 3b), respectively. After reducing at −0.95 V for 30 min, the Raman band around 516 cm$^{-1}$ disappeared due to the reduction of the RuO$_x$ to Ru. Furthermore, only the Ru(0)-H Raman band at a lower wavenumber of 1825 cm$^{-1}$ was observed, This means the band around 1960 cm$^{-1}$ is directly related to the RuO$_x$ surfaces. The Raman band at 715 cm$^{-1}$ shifted to 710 cm$^{-1}$ due to the interactions of adsorbed OH species (*OH) with different valence states Ru surfaces. A similar phenomenon was observed at Ru surfaces during hydrogen oxidation reaction (HOR) in our previous work[37]. DFT calculations confirm that the bending mode of *OH on the top site of the Ru surfaces is 752 cm$^{-1}$ (Supplementary Table 1). Furthermore, we performed ab initio molecular dynamics (AIMD) simulations to explain the existence of *OH at HER potentials and to explore the local structure of *OH at the Ru surfaces. As shown in Supplementary Fig. 10, *OH can be stably adsorbed on the Ru surfaces during AIMD simulations at a very negative potential (−0.96 V vs. SHE). The *OH (on the top site of Ru) accepting two H bonds from neighboring water has a vibrational frequency at 756 cm$^{-1}$, which is interpreted as the bending mode of *OH, and the calculated frequency is very close to the experimental value (710 cm$^{-1}$).

Recently, Dang et al.[38] found good synergistic catalytic effects on Ru/RuO$_2$ interfaces, which can effectively adsorb and dissociate water and have appropriate H and OH binding energies leading to excellent HER electrocatalytic performance. Markovic et al. also proposed a non-covalent interaction mechanism between hydrated cations and covalently bound OH$_{ad}$ to promote the HER[39,40]. Note that the existence of *OH species at Ru surfaces during alkaline HER has rarely been directly observed, although many works speculated that water dissociation is the critical step[29,41–44]. Here, the acquisition of adsorbed *OH on Ru surfaces during alkaline HER provides direct spectral evidence, which is conducive to the in-depth explanation of the mechanism. After oxidizing at +0.95 V for 20 min, the time-course change of the RuO$_2$ Raman peak intensity is shown in Supplementary Fig. 11. The Raman band of RuO$_x$, *OH, and Ru(n+)-H shifted to 523, 716, and 1975 cm$^{-1}$, respectively (Fig. 3b). Consequently, our previous deduction that two distinct *H species are adsorbed on active-sites of Ru surfaces with different valences is supported by further experimental evidence and theoretical simulations.

## HER performance at Ru surfaces in different valence states

Meanwhile, we found an interesting phenomenon after completely reducing and oxidizing the 55 nm Au@2.5 nm Ru catalyst, that is, the oxidized Ru shows superior HER performance than the original and the reduced Ru. The overpotentials of HER at oxidized, original, and reduced Ru surfaces are around 143, 176, and 215 mV at the current density of −10 mA cm$^{-2}$ in a 0.1 M NaOH solution (Fig. 3c). Tafel plots are used to evaluate the kinetics of HER[45]. As shown in Fig. 3d, the corresponding Tafel slopes are 114.8, 133.6, and 146.5 mV dec$^{-1}$, implying the fastest HER kinetics at oxidized Ru surfaces. The differences in the kinetics at different states Ru surfaces could be attributed to either the different valences of surface Ru or the different numbers of active sites on surface Ru. The electrochemical surface area (ECSA) can reflect the active sites on catalyst surfaces, which directly relates to the HER performance[46].

We performed CVs characterizations on 55 nm Au@2.5 nm Ru catalysts in a non-faradaic region (0.10 V–0.20 V) at different scan rates to determine the ECSA. The ECSA study was conducted after the

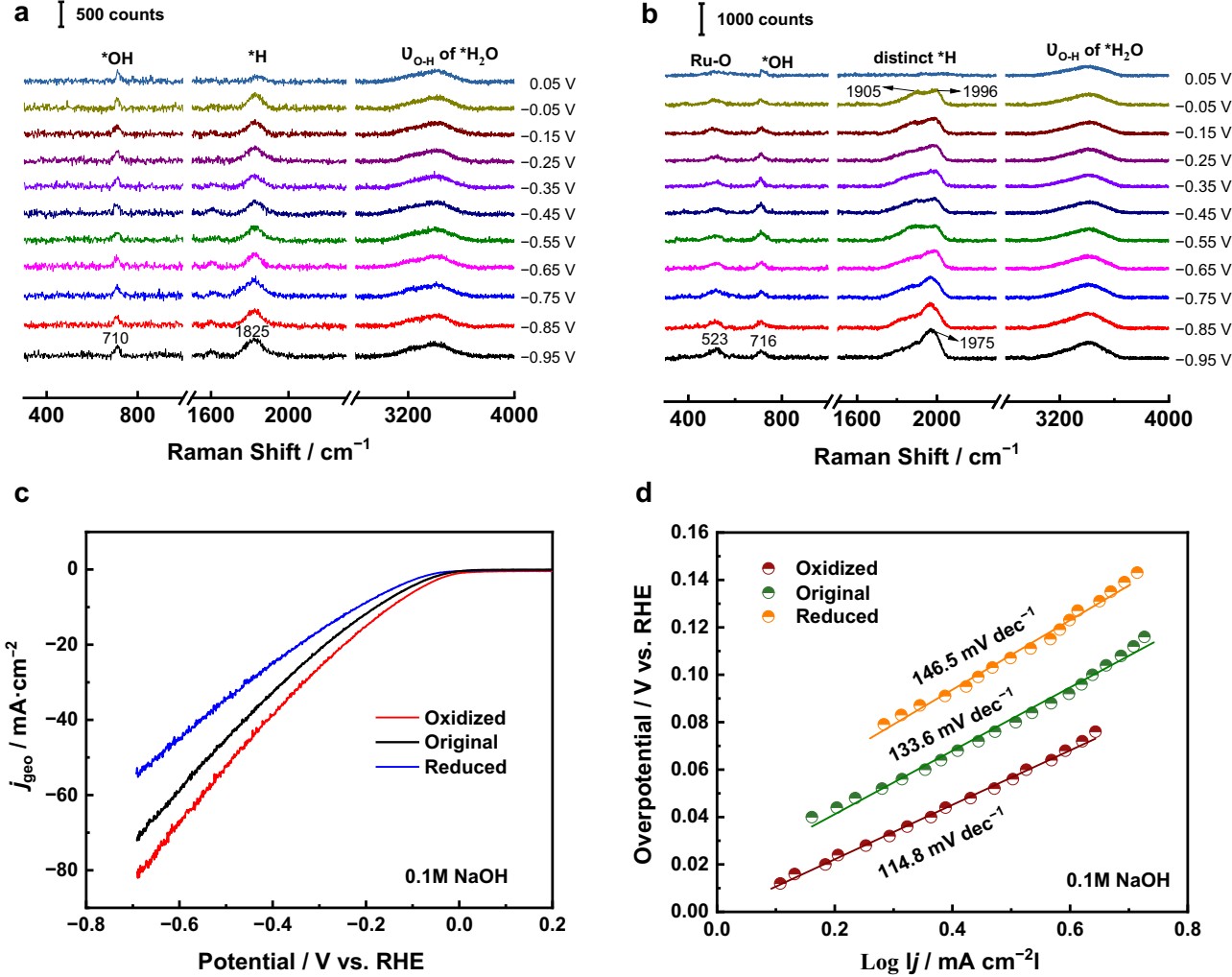

**Fig. 3 | In situ Raman spectra and the corresponding HER performance at Ru surfaces in different valence states.** In situ Raman spectra of the alkaline HER at 55 nm Au@2.5 nm Ru surfaces in 0.1 M NaOH after holding the potential (**a**) at −0.95 V for reduction for 30 mins, and (**b**) at +0.95 V for oxidation for 20 mins.

**c** Polarization curves of the alkaline HER at 55 nm Au@2.5 nm Ru surfaces in different valence states in 0.1 M NaOH (Ar-saturated) with iR correction, 5 mV/s scanning rate, and 1600 r.p.m. **d** Corresponding Tafel plots of the HER polarization curves in (**c**).

electrochemical experiment so that the same amount of catalyst was drop cast. The scan rate dependence of the current densities is shown in Supplementary Fig. 12. The ECSA of original, reduced, and oxidized Ru surfaces are 1236, 1020, and 977 cm², showing the negligible difference in ECSA of the catalysts after the test. Thus, the difference in HER performance could be attributed to the difference in Ru valence states. In the above Linear Sweep Voltammetry (LSV) test, the reduction potential of $RuO_x$ overlaps with the hydrogen evolution potential, resulting in difficulty in separating the contribution of the current. Alternatively, gas chromatography measurements were carried out to quantitatively analyze the hydrogen produced during HER, which can accurately evaluate the HER performance differentiation between Ru(0) and $RuO_x$. The chromatographic test results (Supplementary Fig. 13) showed that the hydrogen yield was the highest catalyzed by the high valence $RuO_x$ (157.1 ppm, 240.0 ppm, and 424.9 ppm for reduced, original, and oxidized Ru respectively), which was consistent with the results shown in electrochemical polarization curve in Fig. 3c that the high-valence $RuO_x$ has the smallest HER overpotential.

**The behavior of interfacial water and its effect on alkaline HER**
Raman signal of interfacial water was observed in our spectra shown in Figs. 2 and 3, which is rare and crucial due to the notorious difficulty of probing it and its key role during the overall alkaline HER process. To

further clarify the effect of interfacial water on the HER performance of Ru catalysts, we performed Gaussian fitting on the broad Raman band of the O-H stretching mode ($v_{O-H}$) of interface water. It can be resolved into three distinct components, corresponding to three types of O–H stretching methods belonging to three types of interfacial water, 4-HB·$H_2O$, 2-HB·$H_2O$, and Na·$H_2O$ respectively[47], on each state of Ru surfaces at each potential (Original state see Fig. 4a, Oxidized, and Reduced states see Supplementary Figs. 14a and 15a). The variation of the vibrational frequency of adsorbate as a function of electrode potential has been attributed to the vibrational Stark effect. The steeper Stark tuning rate shows the higher sensitivity of the adsorbate to the local electric field of the electrode. The Stark tuning rate of $v_{O-H}$ on the investigated Ru surfaces in this work is in the range of 3.6 cm⁻¹/V (Supplementary Fig. 15b, Reduced state, 4-HB·$H_2O$) and 30.2 cm⁻¹/V (Supplementary Fig. 14b, Oxidized state, Na·$H_2O$), which is much smaller than on the surfaces of bare Au (-71 cm⁻¹/V) (Supplementary Fig. 16) and Pd (-76 cm⁻¹/V) NPs[48] but is similar to Pt (-14 cm⁻¹/V)[49]. The lower sensitivity of the interfacial water to the Ru electrode potential compared to the Au and Pd electrodes could be attributed to the indirect contact between them owing to the adsorption of *H on the first layer of the Ru surfaces, while the interfacial water is on the second layer illustrated in Fig. 4b, which is also similar to the Pt electrode surface. We also found that the $v_{Ru-H}$ (1800–2000 cm⁻¹) can be

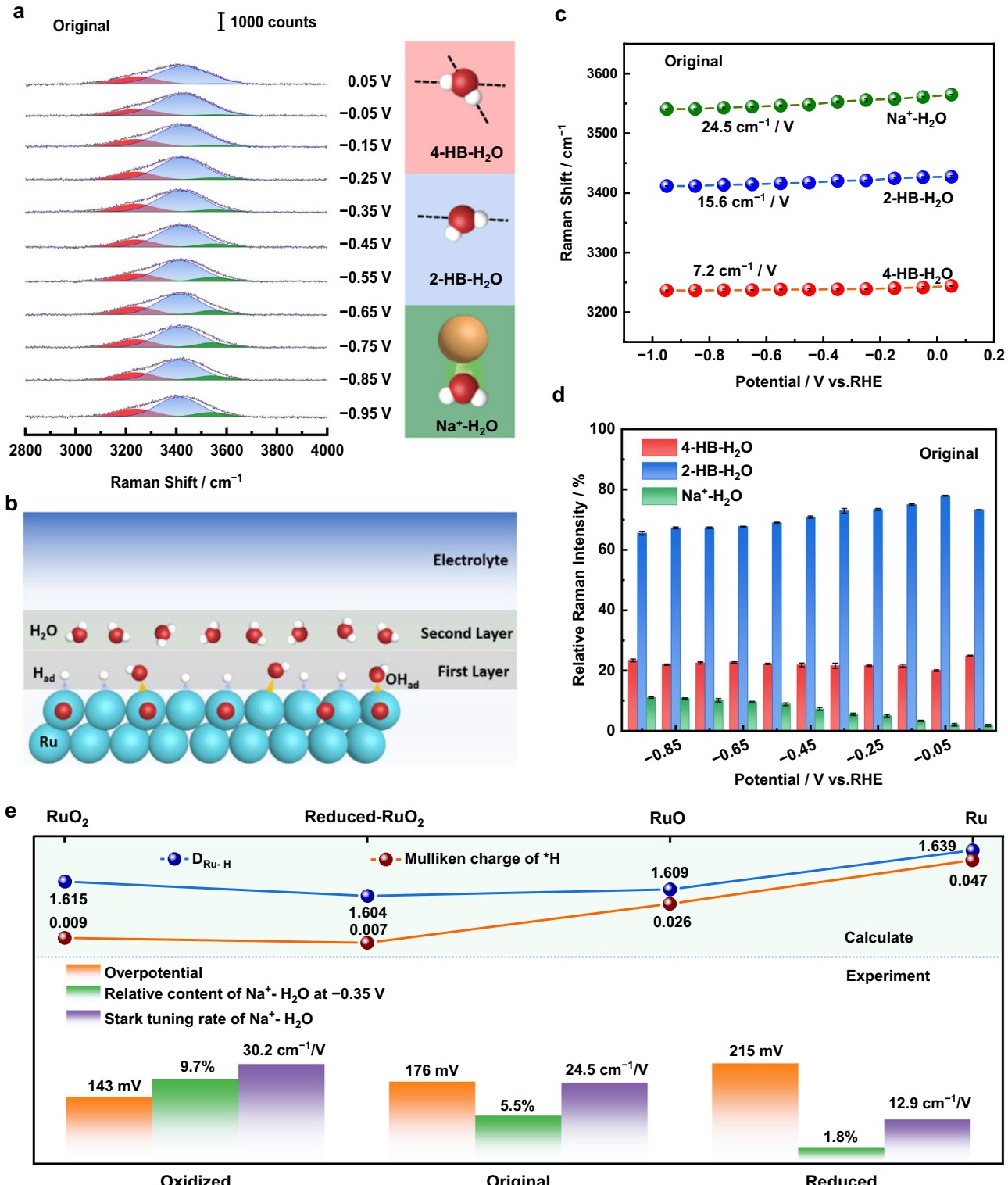

**Fig. 4 | Behavior of interfacial water at 55 nm Au@2.5 nm Ru surfaces and its effect on alkaline HER performance. a** In situ Raman spectra of interfacial water at original Ru surfaces, Gaussian fits of three O–H stretching modes ($\nu_{O-H}$) of 4-coordinated hydrogen-bonded water (4-HB·H$_2$O), 2-coordinated hydrogen-bonded water (2-HB·H$_2$O), and hydrated Na$^+$ ion water (Na·H$_2$O) are shown in red, blue, and green, respectively. **b** Schematic diagram of the states of interfacial species

(interfacial water, *H, and *OH) at Ru surfaces during HER. **c** Frequency plot of changes in the $\nu_{O-H}$ in Raman spectra of interfacial water at original Ru surfaces. **d** Normalized Raman intensities of $\nu_{O-H}$ at original Ru surfaces over HER potentials. **e** The effect of interfacial water and *H on alkaline HER performance. The solution is 0.1 M NaOH. The white, red, orange, and cyan spheres represent H, O, Na, and Ru atoms, respectively.

observed over the HER potential range (Fig. 2b) whereas the $v_{Pd-H}$ could not be observed[47,48]. The probable reason is that H may be rapidly absorbed and embedded into the Pd atomic lattice, resulting in low coverage of H on the Pd surface, which is difficult to be detected[50]. Furthermore, the bending vibration of water can be observed on Pd surfaces at around 1615 cm$^{-1}$ whereas it is very weak on Ru surfaces. These observations further prove the different behavior of *H and interfacial water on Ru and Pd surfaces in the HER potential region.

For Ru at the same surface state, the Stark tuning rate of $v_{O-H}$ is in the order of Na·H$_2$O > 2-HB·H$_2$O > 4-HB·H$_2$O. For instance, the corresponding values of the Stark tuning rate are 24.5, 15.6, and 7.2 cm$^{-1}$/V (Fig. 4c) on the original Ru surfaces. This is owing to the effect of interface electrostatic interaction between the Na·H$_2$O hydrated cation and the Ru surfaces at negative potentials. The Na·H$_2$O, therefore, is closer to the Ru surfaces and is more affected by the electric field of the electrode than the 2-HB·H$_2$O and 4-HB·H$_2$O. For the same type of interfacial water, the oxidized Ru surfaces have the steepest Stark tuning rate of $v_{O-H}$ among the three kinds of Ru surfaces. For instance, the Stark tuning rate of Na·H$_2$O are 12.9 cm$^{-1}$/V (Supplementary Fig. 15b), 24.5 cm$^{-1}$/V (Fig. 4b), and 30.2 cm$^{-1}$/V (Supplementary Fig. 14b) on the reduced, original, and oxidized Ru surfaces. It indicates that the oxidized Ru surfaces have a stronger interaction with interfacial water molecules than the original and reduced Ru surfaces.

Moreover, the vibrational frequency of Na·H$_2$O on RuO$_x$ (3521.1 cm$^{-1}$) (Supplementary Fig. 14b) is lower than that of zero-valent Ru(0) (3563.1 cm$^{-1}$) (Supplementary Fig. 15b). It further shows that the ruthenium oxide surfaces have a stronger interaction with water molecules, which is conducive to break of the O−H bond of adsorbed water[51]. Supplementary Table 2 listed the larger work function of RuO$_2$(110) (5.4 eV) than that of Ru(5.04 eV), which indicates that under the same potential, the adsorbed species on RuO$_x$ surfaces are more negatively charged, resulting in Na·H$_2$O being closer to the surfaces[52]. In addition, the Raman frequency of *H on RuO$_x$ is higher than on Ru. Theoretical calculation based on this shows the shorter Ru-H bond length on the RuO$_x$ than on the Ru surfaces and the smaller *H electronegativity on the RuO$_x$ than on the Ru surfaces (Supplementary Table 2, Fig. 4e). This could be another potential factor in bringing Na·H$_2$O closer to the RuO$_x$ surfaces. The closer of interfacial water to the RuO$_x$ surfaces than the original and reduced surfaces lead to a higher content of Na·H$_2$O on the former (~9.7%) than the latter (~5.5%, and ~1.8%) at the same potential (−0.35 V). The RuO$_x$ surfaces, therefore have superior HER performance than the original and the reduced surfaces shown in Fig. 4e.

Recently, Xu et al.[53] introduced weak hydrogen-bonded water with specific adsorbed organic additives (theophylline derivatives), which increased the intrinsic HOR/HER activity of polycrystalline platinum by 3 times. As mentioned above, Na·H$_2$O here is a much freer form of water. Moreover, the content of 2-HB·H$_2$O increases rapidly near the onset potential of hydrogen evolution (~−0.05 V), while it decreases continuously with the negative potential shift. The content of Na·H$_2$O increases rapidly in the whole potential range (Fig. 4d). We believe that there is a process of converting 2-HB·H$_2$O to Na·H$_2$O due to the closer of the latter to the electrode surfaces than the former, which affects the HER performance as well.

## Insights into alkaline HER mechanism
Herein, we simultaneously obtained dynamic Raman signals of three different interfacial water structures, two distinct *H, and *OH at Ru catalyst surfaces during HER. Recently, Duan et al.[54] combined electrochemical impedance spectroscopy (EIS) experiments and DFT calculations to directly probe the surface and near-surface chemical environment, which reveals the elusive role of alkali metal cations (AM$^+$) in Pt surfaces chemistry and alkaline HER. Comprehensive studies show that cations are not directly attached to the Pt surfaces or

OH$_{ad}$, but are separated by water molecules in the first hydrated shell of the cation, where OH$_{ad}$ in turn acts as an electron-favorable proton acceptor or geometrically favorable proton donor, which together promote the hydrolytic dissociation and Volmer-step kinetics of the Pt surfaces in alkaline media. In the current work, we revealed that *OH is present at Ru surfaces even at very negative potentials. It may serve as an anchoring site for transferring a proton from the interface to the Ru surfaces illustrated in Fig. 5a. In all our modeled surfaces, it is shown that the proton transfer from liquid water to surfaces *OH is energetically favorable, indicating this step is not rate-limiting. For the water dissociation step (Fig. 5b), it is unfavorable to occur on RuO$_2$(110). The free energy is lifted by ~1 eV. In contrast, the energy barrier of the water dissociation step is only ~ 0.1 eV for the reduced RuO$_2$(110) surfaces, and for RuO(100) and Ru(0001) this step is exothermic.

The calculated adsorption-free energies of *H on Ru surfaces in the 2+, 2+ ~ 4+, and 4+ valence states are 0.08, 0.22, and 0.39 eV, respectively. For the last step of HER, H$_2$ formation, all the RuO$_x$ surfaces are exothermic. Thus, *H preferentially desorbs from high-valence Ru surface sites, while this step is hindered on the Ru(0001) surfaces (~−0.5 eV), which is generally known as one of the rate-limiting steps of HER on Ru. Furthermore, different valence states of Ru can effectively regulate the free energy of *H. The RuO$_2$(110) is not probably the reaction site for HER due to the large barrier in water dissociation. Differently, reduced RuO$_2$(110) and RuO(100) exhibit superior HER activity to metallic Ru. It is consistent with the experimental finding that in the presence of RuO$_x$, the apparent HER activity is higher. It is mainly attributed to the proper binding energies of reduced RuO$_2$(110) and RuO(100) for *H$_2$O, *H, and *OH. This work offers insight that synthesized Ru with higher valence states will be favorable for HER. Fortunately, some recent work report various strategies for stabilizing metals in high-valence states[55–57].

To summarize, we employed in situ Raman spectroscopy to monitor the alkaline HER process at Ru catalyst surfaces using 55 nm Au@2.5 nm Ru NPs. Combining the simultaneously obtained direct and dynamic spectral evidence of interfacial water, two distinct *H, and *OH at different valences Ru surfaces with the DFT calculation results, we outlined a detailed schematic for the alkaline HER mechanism at Ru surfaces. It includes the dissociation of interfacial water, the adsorption of *H and *OH intermediates, and their interactions with Ru surfaces in different valence states. Our findings show the crucial role of interfacial water, two distinct *H and *OH intermediates at Ru catalyst surfaces during the alkaline HER process. The local cation tuning effect of Na·H$_2$O and the large work function of high-valence Ru(n+) surfaces induce a stronger interaction, thus promoting interfacial water dissociation. Furthermore, the high-valence Ru(n+) surfaces have more moderate adsorption energies for interfacial water, *H, and *OH during HER. Last but not least, OH$_{ad}$ acting as an electron-favorable proton acceptor or geometrically favorable proton donor promotes the interfacial water dissociation and Volmer-step kinetics involving Ru(n+)-H on the RuO$_x$ surfaces in alkaline media. They, therefore, facilitate the alkaline HER performance. Our findings demonstrate a mechanism of valence state tuning effect on interfacial water dissociation and key intermediate species with catalytic activity improvement, which provide valuable insights for the design and synthesis of highly efficient catalysts.

## Methods
### Preparation of 55 nm Au nanoparticles (NPs)
55 nm Au seeds were synthesized according to Frens'method[58]. In brief, 200 mL of 0.01% HAuCl$_4$ aqueous solution was heated to boil. 1.5 mL of 1% trisodium citrate solution was quickly added under stirring and kept boiling for 30 min. Afterward, the final solution was cooled down to room temperature for later use.

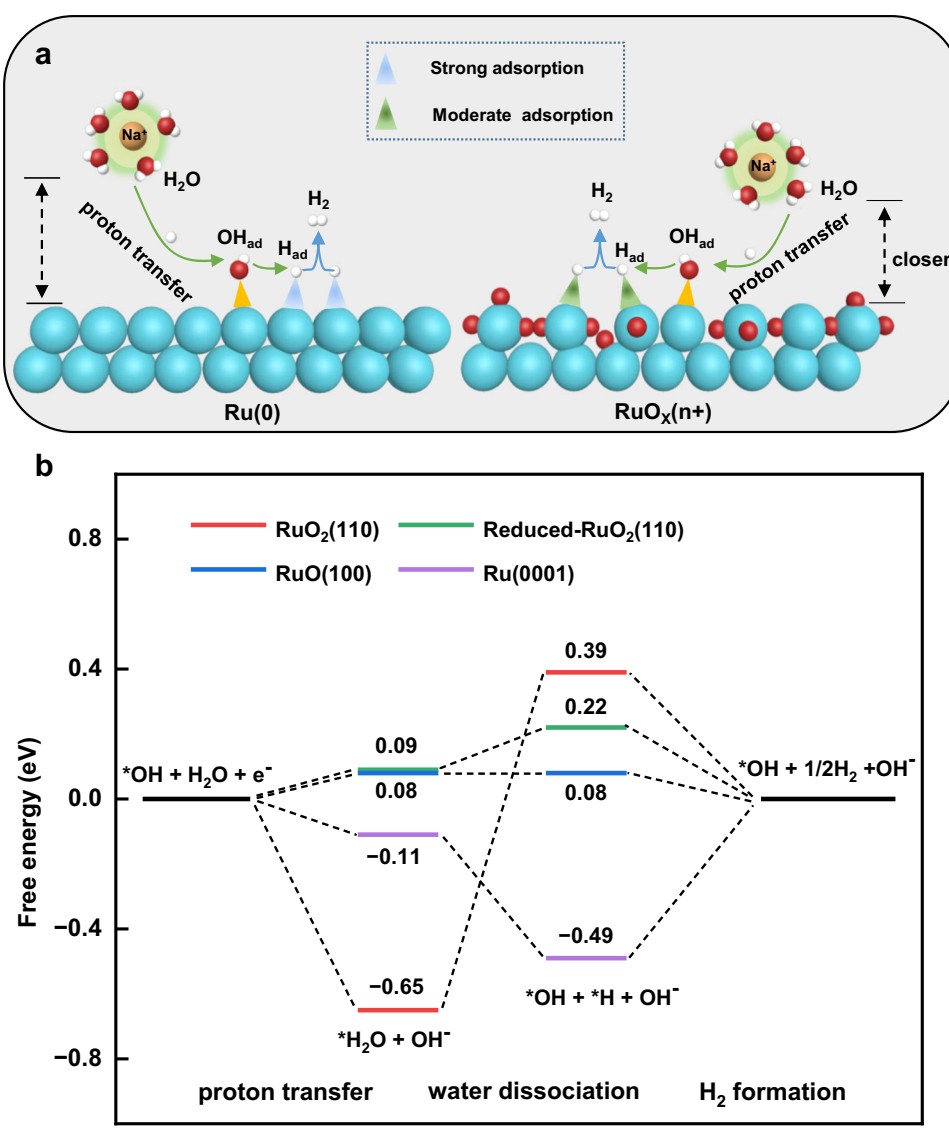

**Fig. 5 | Alkaline HER mechanism at Ru surfaces in different valence states. a** Schematic diagram of HER at Ru(0) and RuO$_x$(n+) surfaces. **b** Step-wise free energy profiles of HER at Ru surfaces in different valence states.

### Preparation of Au@Ru core-shell NPs

30 mL of as-prepared 55 nm Au nanoparticle sol was heated and stirred to mix uniformly. After heating to boil, 1 mL of 2 mM RuCl$_3$ (pH = 2) solution and 1 mL of 1% trisodium citrate solution were added simultaneously and continuously stirred for 30 min after dripping, and get Au@Ru core-shell nanoparticles.

### Preparation of Au@Ru catalyst ink

Inductively coupled plasma atomic emission spectroscopy (ICP-AES) measurements show the concentrations of Au and Ru in the Au@Ru colloidal solutions in Supplementary Table 3. The Au@Ru colloidal solution was condensed to 1.5 mg mL$^{-1}$ (relative to Ru). 50 μL of condensed Au@Ru sol was mixed with 175 μL isopropanol and 25 μL Nafion (2 wt%). 0.75 mg of Vulcan XC-72R (Cabot Co.) was then added to the solution and sonicated for 30 min (Vulcan XC acted as structural support for the Au@Ru NPs). For the Au@Ru/C catalyst ink, a Ru loading of 0.3 mg mL$^{-1}$ was used. The carbon content of catalyst ink was also studied to determine its impact on HER activity. 50 μL of the catalyst ink was then drop-cast onto the surface of the carbon cloth and dried naturally before electrochemical measurements.

### Electrochemical measurements

Electrochemical tests were performed on a CHI 760E electrochemical workstation. As mentioned above, the Au@Ru catalyst was decorated on the surface of the treated carbon cloth working electrode, with a graphite rod counter electrode, and an RHE reference electrode. A fresh solution was used as the electrolyte for each electrochemical test. Before starting the HER electrochemical measurements, the solution was deoxygenated with Ar and stirred during the test, and the surface was protected by an Ar stream. The potential was swept at a scan rate of 5 mV s$^{-1}$ from 0.2 V to −0.7 V (vs. RHE), and all HER polarization curves were current-resistance (iR) corrected.

### In situ electrochemical Raman measurements

Glassy carbon (GC, D 0.2 cm) working electrodes were polished with 1, 0.3, and 0.05 μm Al$_2$O$_3$ powder slurries to obtain a smooth surface, then washed with ultrapure Mili-Q (18.2 MΩ cm) water, 0.5 M H$_2$SO$_4$, and ethanol before sonication, repeated 3 times. Finally, the surface was blown dry with N$_2$. Then, 10 μL of catalyst sol was drop-cast onto the surface of the GC electrode and dried naturally. In situ Raman spectroscopic experiments were performed with a confocal

microscope Raman system Xplora Plus (HORIBA France). The excitation wavelength of the semiconductor laser was 638 nm, the laser power was 2.8 mW, and all Raman measurements were performed with a 50× microscope objective with a numerical aperture of 0.55. Raman frequencies were calibrated using silicon wafers during each experiment. In situ electrochemical Raman experiments were carried out in an in-house made Raman cell with Au@Ru nanocatalyst decorated GC electrode as the working electrode, Pt wire as a counter electrode, and either a Hg/HgO, or SCE electrode as a reference electrode. An Autolab PGSTAT30 (Metrohm) potentiostat was used to control the potential.

### Theoretical calculation details

All the DFT calculations were carried out by the CP2K package. Goedecker-Teter-Hutter (GTH) pseudopotentials were used to represent the core electrons. The Gaussian basis set was double-$\zeta$ with one set of polarisation functions (DZVP), and the energy cutoff was set to 400 Ry. We employed PBE functional[59] to describe the exchange-correlation effects, and the dispersion correction was applied in all calculations with the Grimme D3 method[60].

Metallic Ru and three kinds of Ru-RuO$_x$ (they differed in the valence states of Ru in oxide) were studied in this work, as shown in Fig. 2d. The Ru(0001) surface was modeled by a 6 × 6 slab with four atomic layers. The Ru-RuO$_x$ surfaces were constructed by putting RuO$_x$ on top of Ru(0001), and keeping the lattice mismatch between Ru and RuO$_x$ lower than 5%. RuO$_2$(110) and RuO(100) were chosen for this study, and the valence states of their Ru are +4 and +2, respectively. As no stable bulk RuO was reported, the lattice of RuO was optimized based on NiO. We also considered a reduced RuO$_2$(110) surface, for which the surface bridge O atoms were leached. By analyzing the Bader charge, it was found that the Bader charge of the 5c-Ru was decreased from +1.52 e$_0$ to +1.39 e$_0$ by removing the surface O of RuO$_2$(110), indicating the valence states of the 5c-Ru were decreased from +4 to ~+3.7. Due to the large unit cells modeled in this work, the studied surfaces were computed only on the Γ point.

The adsorption of H and OH was computed on different surfaces, and their calculated vibrational frequencies were listed in Supplementary Table 1. The reaction pathways of the hydrogen evolution were plotted based on the computational hydrogen electrode (CHE) scheme proposed by Norskov and co-workers[61]. The Gibbs free energy (G) was calculated with $G = E + ZPE - TS + E_{sol}$, where E, ZPE, TS, and E$_{sol}$ denote the total energy, zero-point energy, entropy, and solvation free energy, respectively. The solvation effect of adsorbed H, OH, and H$_2$O was taken into account with correction values as listed in Supplementary Table 4.

An electrified Ru(0001)/water interface was modeled (see Supplementary Fig. 10), in which the Na$^+$ ions fully compensated the surface charges at interfaces. The model was computed with second-generation Car-Parrinello molecular dynamics (SGCPMD). The correction step was obtained by 5 iterations of the orbital transformation (OT) optimization, and the integration time for each step was 0.5 fs. The target temperature was set to 330 K. The Langevin friction coefficient ($\gamma$_L) was set to 0.001 fs$^{-1}$, and the intrinsic friction coefficients ($\gamma_D$) were 3 × 10$^{-4}$ fs$^{-1}$ for H$_2$O and ions, and 6 × 10$^{-5}$ fs$^{-1}$ for Ru. The bottom two Ru layers and Ne layers were fixed during MD simulations. 1 ps (2000 steps) of the trajectory was used to pre-equilibrate the system, then followed by a production period of 9 ps.

### Data availability

The data that support the findings of this study have been included in the main text and Supplementary Information. All other relevant data supporting the findings of this study are available from the corresponding authors upon request.

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

## Acknowledgements

This work was supported by grants from the National Key Research and Development Program of China (2020YFB1505800) (J.-C.D.), the National Natural and Science Foundation of China (21925404 (J.-F.L.), 22222903 (J.-C.D.), 52271229 (J.-C.D.), 22021001 (J.-F.L.), 22005130 (X.-M.L.), 22272069 (X.-M.L.), 22272193 (J.-B.L.), 21991151 (J.-C.D.), and 21902136 (J.-B.L.)), and the Fundamental Research Funds for the Central Universities (20720210043 (J.-C.D.)), and Yongjiang Talent Introduction Program (2021A-115-G) (J.-B.L.).

## Author contributions

J.-C.D. and J.-F.L. directed the project. X.C. and X.-T.W. performed the main experimental works. J.-B.L. and X.W. performed the theoretical calculation works. S.-M.L., Y.-J.Z., Y.-H.W., and Y.Z performed the material preparation and participated in some experimental work. X.C., X.-T.W., J.-B.L., X.-M.L., and P.R. analyzed the data and wrote the manuscript. All the authors discussed the results and commented on the manuscript.

## Competing interests

The authors declare no competing interests
