## [Peer Review File · Nature Communications]

Revealing the role of interfacial water and key intermediates at ruthenium surfaces in the alkaline hydrogen evolution reactionREVIEWER COMMENTS

Reviewer #1 (Remarks to the Author):

The authors have used a Au-nanoparticle of about 55 nm, which was coated with a HER active Ru phase of about 2 nm in thickness to investigate the process of HER with Raman spectroscopy. Due to the use of this combination of Au-Ru one can call this core-shell nanoparticle-enhanced Raman spectroscopy, which has been explored in the past years to investigate surface processes in the field of catalysis, and here it is now applied to better understand what is happening at the surface of Ru and if one can determine *H and *OH species as intermediates. The authors have made use of D2O to further substantiate their findings, and they also - backed with theoretical calculations - propose interfacial water interactions and changes when different potentials are applied.

I have read the paper with interest and as we are also working in this field of research, thereby using e.g. Raman and IR spectroscopy, I have been interested in both the spectral assignments as well as in the proper use and characterisation of the Au-Ru nanoparticle structure to provide the necessary Raman measuring capabilities. Here, I feel that the authors are overly confident that they have made a pinhole-free structure and that they do not have any contributions from other parts of their "measurement probing system". I believe that a proper characterization is needed before it becomes clear to me that the system is indeed a nice overlay of Ru fully surrounding the Au nanoparticle. Furthermore, what is the distance that the enhancement of the Raman signals by Au are sufficiently guaranteed; and what happens when the shell thickness is changing; as well as the need to have e.g. two Au nanoparticles close to each other and having interfacial structures in between needed to make sure that the signals can be analysed. I also believe that the D2O experiments are not sufficiently explained, and more data have to be incorporated to make the proper analysis; hence experiments as in Figure 2b have to be done, but now with D2O. I am doubtful about some of the assignments and shifts explanations provided by the authors; also here more backing and clarifications are needed to better understand what is now what; and how sure we can be about the assignments (and related shifts).

Summarizing, although of interest and also having novelty, I believe that the work needs some careful evaluation before it can be published in a journal such as Nature Communications. I am not yet convinced that the observations made are indeed due to the surface species proposed by the authors.

Reviewer #2 (Remarks to the Author):

This manuscript reports on in situ plasmon-enhanced Raman spectroscopy of alkaline HER process at Ru catalyst surfaces. *H and *OH intermediates were directly detected on Ru surfaces and the importance of the valence states of Ru was discussed by assistance of the DFT simulations. This work provides the microscopic outline of the hydrogen evolution process with the strong evidence, and thus will be suitable for publication after revisions noted.

- The Stark tuning rate of OH stretch was smaller on Ru surface than on Au or Pd surface. This difference was explained by the indirect contact between Ru surface and water molecules as a presence of the adsorbed *H on Ru surface. However, the adsorbed *H should similarly exist on Pd surface in the HER potential region, which seems to be inconsistent with the explanation.

- For the peak separation analysis of the OH stretch band, error bars should be indicated because the baseline subtraction may affect the intensities of Na+H2O and 4-HB-H2O.

- The original Ru surfaces are partially oxidized as shown in XPS spectra in Supplementary Fig. 4. This information should be presented before discussing the potential dependence of Raman spectra in Fig. 2. Otherwise, the potential dependence of Ru-H band for different Ru valence states is very confusing.

Actually, the series of Raman spectra shown in Fig. 2 should include not only the potential-induced variations but also the time-course changes. This should be clearly discussed.

Reviewer #3 (Remarks to the Author):

Comments on the Manuscript: NCOMMS-23-10848-T

The present work presents a deep insight into the HER mechanism on Ru surfaces in alkaline media. By using Au@Ru core-shell nanoparticles, key spectroscopic information was obtained from hydrogen adsorption (H^*), hydroxide adsorption (OH^-) and water dissociation process. This experimental data combined with theoretical calculation revealed the role of different valence Ru on the surface (+2, +2 - +4, + 4) in the adsorption of H, OH and in the whole HER mechanism, concluding that OH^- adsorbed on RuO_2 serve as an anchoring that helps transferring protons from the interface to the Ru surfaces, favouring the H^* recombination and leading to a higher hydrogen evolution. Findings from Raman signals help to support this observation by funding OH^* signal on RuO_2 surfaces at large negative potentials and two distinct H^* . Overall, the work was well executed, and the experimental design was very well planned with a solid theoretical calculation to support the experimental observations. Despite that, I have some questions and concerns about the work that I would like to clarify. After this, I believe the manuscript needs some minor revisions to address the above comments, prior it could be considered for publication in Nature Communications.

1. Why do the authors perform the experiment in this media and not in a more tested typical electrolytic conditions such as 1 M KOH? Authors are encouraged to provide a solid reason for this. In any case, have the authors tried performing the experiment in such conditions?
2. In Figure 2b. As far as I understood, potential was scanned from more negative to more positive potentials, have the authors try performing the experiment in the opposite direction? If yes, could the authors comment something on this? Could be expected differences depending on the direction in which the experiment was performed?
3. Regarding the isotopic labelling experiment, Do the authors only perform the experiment at -0.35 V? How about other relevant potentials? For instance, what's the comparison with the behaviour of the signals at a more positive potential, i.e., closer to the onset potential? Is there a different behaviour? or the comparison at a more negative potential, i.e., -0.8 V. It would be interesting to have this comparison with the isotopic labelling experiment, so a deeper generalization could be extracted.
4. In Supplementary Figure 4, the authors are encouraged to present the O XPS signal as other reports used to present, so a better and accurate correlation in the types of different valences Ru can be extracted.
5. Authors claims that "With potential increasing, the vibrational frequency of the $Ru(n+)$ -H Raman band (red square) around 1960 cm^{-1} red-shifts and its intensity (blue square) gradually decreases, which could be due to the gradual reduction of the high valence $Ru(n+)$ into zero-valent $Ru(0)$ under the reduction potential". I have some serious concern about this expression and about interpretation of the data. I would expect this behaviour to be more pronounced at more negative potentials, not at more positive one as the author show in their results. In general, at more negative potential I expect that Ru in more in the form of $Ru(0)$ than in the form of a high valence Ru. Could the authors explain this fact?
6. Regarding Figure 2b and related to the previous comment. It would be interesting to see how the evolution of the band corresponding to Ru-O (516 cm^{-1}) with the potential, plotted in a similar way that has been done for H signals. The representation of this evolution will give a better understanding about the composition of the different valence Ru in the surface as the potential is being scanned.
7. In the expression "We performed CVs characterizations on Au@Ru catalysts in a non-faradaic area at different scan rates to determine the ECSA" change non-faradaic area by "non-faradaic region".
8. I guess the ECSA studied was conducted after the electrochemical experiment, so that the same amount of catalyst was drop casted... if it was like this, I encourage to the author to mention in the text so the reader could clearly relate the area with the actual LSV experiment shown for HER with the corresponding ECSA.

9. Regarding ECSA calculations, for me is confusing that the authors oxidize the electrode at +0.95 V, however the study of the ECSA was done in a range between +0.80 V and + 1.0 V. Maybe the authors made a mistake in the potential region. If not, the authors should explain why using this potential value.

Reviewer #1 (Remarks to the Author):

The authors have used an Au-nanoparticle of about 55 nm, which was coated with a HER active Ru phase of about 2 nm in thickness to investigate the process of HER with Raman spectroscopy. Due to the use of this combination of Au-Ru, one can call this core-shell nanoparticle-enhanced Raman spectroscopy, which has been explored in the past years to investigate surface processes in the field of catalysis, and here it is now applied to better understand what is happening at the surface of Ru and if one can determine *H and *OH species as intermediates. The authors have made use of D₂O to further substantiate their findings, and they also - backed with theoretical calculations - propose interfacial water interactions and changes when different potentials are applied.

I have read the paper with interest and as we are also working in this field of research, thereby using e.g. Raman and IR spectroscopy, I have been interested in both the spectral assignments as well as in the proper use and characterization of the Au-Ru nanoparticle structure to provide the necessary Raman measuring capabilities. Here, I feel that the authors are overly confident that they have made a pinhole-free structure and that they do not have any contributions from other parts of their "measurement probing system". I believe that a proper characterization is needed before it becomes clear to me that the system is indeed a nice overlay of Ru fully surrounding the Au nanoparticle. Furthermore, what is the distance that the enhancement of the Raman signals by Au is sufficiently guaranteed; and what happens when the shell thickness is changing; as well as the need to have e.g. two Au nanoparticles close to each other and interfacial structures in between needed to make sure that the signals can be analyzed. I also believe that the D₂O experiments are not sufficiently explained, and more data have to be incorporated to make the proper analysis; hence experiments as in Figure 2b have to be done, but now with D₂O. I am doubtful about some of the assignments and shifts explanations provided by the authors; also here more backing and clarifications are needed to better understand what is now; and how sure we can be about the assignments (and related shifts).

Summarizing, although of interest and also novelty, I believe that the work needs some careful evaluation before it can be published in a journal such as Nature Communications. I am not yet convinced that the observations made are indeed due to the surface species proposed by the authors.

Response: We thank the referee for the insightful and valuable comments on our manuscript. Following the suggestion of the referee, we have either carried out new experiments or cited reference papers to explain further and complete our points. Here are point-by-point answers to his/her comments, with the corresponding changes highlighted in yellow in the revised manuscript and supplementary information (SI).

Comment 1: Here, I feel that the authors are overly confident that they have made a pinhole-free structure and that they do not have any contributions from other parts of their "measurement probing system". I believe that a proper characterization is needed before it becomes clear to me that the system is indeed a nice overlay of Ru fully surrounding the Au nanoparticle.

Response 1: We thank the referee for the valuable comments. The coating of the shell layer of the core-shell nanoparticles (NPs) becomes more compact with increasing the shell thickness. Generally, the pinhole-free structure can be made when the shell is thicker than 2.0 nm (Chem. Rev. 2017, 117, 5002-5069). The Au@Ru NPs used in our HER investigations are pinhole-free, which will be demonstrated from three aspects (morphology, CV curve, and Raman spectra) as we presented in Au@Pt, Au@Pd, Au@SiO₂, and Ag@SiO₂ NPs in our previous publications (Nature 2010, 464 392-395; J. Phys. Chem. C 2016, 120, 20684-20691; Langmuir 2006, 22, 10372-10379): (1) The SEM and TEM images of the used Au@Ru NPs in Figure R1 (a) and (b) (Supplementary Figure 1 (a) and (b)) show a uniform size and good dispersibility with a diameter of the Au core of around 55 nm and a thickness of the Ru shell of around 2.5 nm and no obvious pinholes can be observed. (2) In Figure R1 (c) (Supplementary Figure 2 (a)), the CV curve of 55 nm Au NPs (red) shows two Au reduction peaks at around 0.8 and 1.4 V (Angew. Chem. Int. Ed. 2021, 60, 13452-13462; J. Am. Chem.

Soc. 2015, 137, 7648-7651; Trans. IMF 2007, 85, 194-201). The CV curve of 55 nm Au@2.5 nm Ru NPs (blue) shows two pairs of redox peaks at around 0.17/0.43 V and 0.67/1.04 V attributed to the Ru(+3)/Ru(+4) and Ru(+4)/Ru(+6) (J. Electroanal. Chem. 2021, 881, 114955; Electrochim. Acta 2020, 354, 136625; Trans. IMF 2007, 85, 194-201;) whereas no Au reduction peaks appear, which indicates that the NPs are pinhole-free. (3) The electrochemical method provides a piece of general macroscopic information. If the pinholes occupy an extremely small proportion of the surface, the generated current in the electrochemical CV would be too small to be observed. So, we used a more direct and sensitive pinhole detection method - Core-shell nanoparticle-enhanced Raman spectroscopy. Since the Raman signal generated on gold is several orders of magnitude stronger than that on transition metal, once probe molecules are adsorbed on the gold core through pinholes, a strong enhanced Raman signal will be generated. We have carried out new Raman measurements to confirm further the pinhole-free structure of the 55 nm Au@2.5 nm Ru NPs using CO as a probe molecule.

As can be observed in Figure R1 (d) (Supplementary Figure 2 (b)), the Raman spectrum of CO adsorbed on 55 nm Au@0.7 nm Ru NPs (red) at 0.05 V shows two Raman bands at around 2005 cm^{-1} and 2125 cm^{-1} . They are ascribed to $\nu_{\text{CO}(\text{top})}$ on Ru and $\nu_{\text{CO}(\text{top})}$ on Au, respectively (Angew. Chem. Int. Ed. 2021, 60, 13452-13462; J. Am. Chem. Soc. 2003, 125, 2282-2290; J. Phys. Chem. C 2016, 120, 20684-20691; J. Korean Electrochem. Soc. 2002, 5, 221-225). The Ru shell is filled with pinholes, which results in the probed CO molecule coming into contact with the Au core; the Raman band of $\nu_{\text{CO}(\text{top})}$ on Au, therefore, can be observed. Nevertheless, the Raman spectrum of CO adsorbed on 55 nm Au@2.5 nm Ru NPs (black) shows only one Raman band of $\nu_{\text{CO}(\text{top})}$ on Ru at around 2010 cm^{-1} at 0.05 V, and it blueshifts to around 2019 cm^{-1} at 0.25 V due to the Stark-tuning effect of the electrode. The absence of the Raman band of $\nu_{\text{CO}(\text{top})}$ on Au again demonstrates the pinhole-free structure of the used 55 nm Au@2.5 nm Ru NPs.

We believe that we have provided sufficient and solid evidence to prove that our used 55 nm Au@2.5 nm Ru NPs are indeed a nice overlay of Ru fully surrounding the

Au nanoparticle pinhole-free structures as the referee requested, which can eliminate the contributions from other parts of our "measurement probing system". Figure R1 (c) and (d) have been added as Supplementary Figures 2 (a) and (b), and the related detailed explanation has been added and highlighted as well in SI. A concise explanation has been added and highlighted in "Construction and characterization of Au@Ru core-shell nanoparticles (NPs)" in the revised manuscript due to the word limit of the journal.

Figure R1 (a) SEM image and (b) TEM image of 55 nm Au@2.5 nm Ru NPs. (c) Comparison of cyclic voltammogram curves of 55 nm Au NPs (red) and 55 nm Au@2.5 nm Ru NPs (blue) in a solution of 0.1M NaClO₄. (d) enhanced Raman spectra of CO adsorbed on 55 nm Au@0.7 nm Ru NPs (red, with pinholes) and 55 nm Au@2.5 nm Ru NPs (black, pinhole-free) in a solution of 0.1 M NaClO₄ saturated by CO.

Comment 2: Furthermore, what is the distance that the enhancement of the Raman signals by Au is sufficiently guaranteed; and what happens when the shell thickness is changing; as well as the need to have e.g. two Au nanoparticles close to each other and interfacial structures in between needed to make sure that the signals can be analyzed.

Response 2: We thank the referee for his/her expertise. We agree with the referee that the thickness of the shell and the interparticle distance influence the Raman enhancement effect. Although the phenomenon that enhanced Raman spectral intensity of the adsorbed molecules decreases with increasing shell thickness has been reported in our other systems like Au@Pt, Au@Pd, and Au@SiO₂ (Nature 2010, 464 392-395; Langmuir 2006, 22, 10372-10379; J. Phys. Chem. C 2016, 120, 20684-20691; J. Raman Spectrosc. 2015, 46, 1200-1204) here, we try to depict the relationship between the Ru shell thickness and the enhanced Raman spectral intensity of the Au@Ru NPs experimentally and theoretically.

Figure R2 (a) shows the TEM images of the Au@Ru NPs with different Ru shell thicknesses of 0.7, 1.4, 2.5, 5.0, and 7.0 nm, and Figure R2 (b) are the corresponding enhanced Raman spectra of CO adsorbed on these Au@Ru NPs. The intensity of the $\nu_{\text{CO(top)}}$ decreases dramatically as the interparticle distance controlled by the Ru shell thickness increases due to the decay of the plasmon coupling efficiency between particles. To make it more obvious, the intensity of the $\nu_{\text{CO(top)}}$ Raman band against Ru shell thickness was plotted (blue) and simulated by 3D-Finite-difference time-domain (3D-FDTD) (red) shown in Figure R2 (c). Our 3D-FDTD calculations prove that the corresponding highest enhancement of the Raman signal is 7×10^6 for the 2.5 nm shell. The strong enhancement effect of the electric field significantly amplifies the Raman signal of the species in our systems. They are therefore detected. Additionally, it can be observed in Figure R2 (b) that the Raman band of $\nu_{\text{CO(top)}}$ at 55 nm Au@ 0.7 nm Ru is at 2005 cm^{-1} , which redshifts by 5 cm^{-1} on 55 nm Au@1.4 nm Ru and keeps consistent with increasing of Ru shell thickness till 7.0 nm. This indicates that the interaction between the CO and the Ru surfaces is influenced by the Au core only when the Ru thickness is thinner than 1.4 nm. Namely, the Au@Ru NPs are pinhole-free when the Ru thickness is thicker than 1.4 nm. Both experimental and theoretical results show the exponential decrease of the Raman intensity with increasing the shell thickness. Synergistically considering the pinhole-free structure and the Ru shell thickness-enhancement effect relationship, we selected the 55 nm Au@2.5 nm Ru NPs for our

HER investigation. Figure R2 has been added as Supplementary Figure 3 and the related detailed explanation has been added and highlighted as well in SI. The concise explanation has been mentioned and highlighted in “Construction and characterization of Au@Ru core-shell nanoparticles (NPs)” in the revised manuscript.

Figure R2 (a) TEM images of the Au@Ru NPs with different Ru shell thicknesses (0.7, 1.4, 2.5, 5.0, and 7.0 nm from the left to the right). (b) Corresponding enhanced Raman spectra of CO adsorbed on different thicknesses Ru shell Au@Ru NPs shown in (a) in a 0.1 M NaClO₄ solution saturated with CO at 0.05 V. (c) The shell thickness dependence of the normalized enhanced Raman spectral intensity of CO adsorbed on different Ru shell thicknesses Au@Ru NPs (blue) and the corresponding 3D-FDTD calculation result (red). Inset is FDTD simulation of the electric field distribution on the surface of 55 nm Au@2.5 nm Ru dimer.

Comment 3: I also believe that the D₂O experiments are not sufficiently explained, and more data have to be incorporated to make the proper analysis; hence experiments as in Figure 2b have to be done, but now with D₂O. I am doubtful about some of the assignments and shifts explanations provided by the authors; also here more backing and clarifications are needed to better understand what is now what; and how sure we

can be about the assignments (and related shifts).

Response 3: We have performed in situ D₂O experiments in the HER potential range from -0.95 V to +0.05 V shown in Figure R3 and Supplementary Figure 6 as H₂O experiment in Figure 2b to provide complete data for a sufficient explanation, as the referee suggested. Generally, except for the corresponding displacement of the specific peak associated with 'H', the changing trend of the Raman spectra in the D₂O experiment is similar to that in H₂O. For example, in the isotope experiment, we also observed the Raman vibration signals of Ru-O (516 cm⁻¹), Ru-OD (672 cm⁻¹), two distinct D (1318 cm⁻¹, 1411 cm⁻¹), and interfacial water D₂O (2503 cm⁻¹) on Ru at -0.95V. The shifts of these vibrational frequencies accord with the expected shifts from the mass conversion of the formula ($\gamma=71.1\%$) (as calculated, see mass formula section of Supplementary Information) and previous reports in the literature (Nat. Catal. 2021, 4, 711-718; J. Am. Chem. Soc. 2020, 142, 8748-8754). With the potential shifts positively, the intensity of *D on the high-valence Ru (1411 cm⁻¹) gradually decreases whereas the frequency of *D on zero-valence Ru (1318 cm⁻¹) redshifts. Our isotope experiment further verifies the phenomenon observed in the H₂O experiment.

Regarding the assignment of the Raman bands and their shifts, except the performed deuterium isotopic substitution experiment, they are further verified by mass conversion formula, reference papers, and DFT calculations as we did in our previous publications (Nat. Energy 2019, 4, 60-67; J. Am. Chem. Soc. 2019, 142, 715-719). These contents have been presented in the manuscript as follows: A deuterium isotopic experiment was performed (Supplementary Fig. 6), and potential at -0.35 V was selected to verify the species observed in Fig. 2b. Substituting D₂O for H₂O, the Raman band around 516 cm⁻¹ remains stable, indicating that this species is not related to an 'H' atom and could be attributed to RuO_x surface oxide originating from the Ru catalysts. Nevertheless, the Raman bands around 715 cm⁻¹, 1827 cm⁻¹, 1950 cm⁻¹, and 3420 cm⁻¹ shift to around 674 cm⁻¹, 1328 cm⁻¹, 1402 cm⁻¹, and 2502 cm⁻¹, respectively, which imply that these species include the 'H' atom. The shifts of these vibrational frequencies accord with the expected shifts from the mass conversion of the formula ($\gamma=71.1\%$) (as

calculated, see mass formula section of Supplementary Information) and previous reports in the literature (Nat. Catal. 2021, 4, 711-718; J. Am. Chem. Soc. 2020, 142, 8748-8754). The Raman bands around 1827 cm^{-1} and 3420 cm^{-1} can be attributed to the vibrations of the Ru-H bond of adsorbed hydrogen (*H) and interfacial water at the metallic Ru(0) surfaces, respectively (Nat. Catal. 2021, 4, 711-718; J. Am. Chem. Soc. 2020, 142, 8748-8754). To the best of our knowledge, the Raman band around 1950 cm^{-1} at Ru surfaces during HER has not been reported, which was preliminarily attributed to the Ru-H bond of adsorbed hydrogen (*H) at oxidized Ru(n+) surfaces. The oxidation of the most out layer of the Ru shell has been demonstrated by XPS spectra in Supplementary Fig. 4. The XPS spectrum of the Ru 3p electrons in (a) shows two characteristic peaks around 484.7 eV and 462.3 eV, corresponding to metallic Ru(0), and the other two peaks around 486.6 eV and 464.2 eV correspond to oxidized Ru(n+). The O 1s XPS spectrum for Au@Ru NPs can be assigned to lattice oxygen, adsorbed hydroxide, and water ($\text{OH}^-/\text{H}_2\text{O}$) located at 530.4 and 532.5 eV, respectively (Electrochem. Commun. 2007, 9, 239-244, J. Energy Chem. 2021, 54, 510-518), which demonstrates the oxidation of the most out-layer of the Ru shell to $\text{RuO}_2(4+)$ under atmospheric conditions. Therefore, different Ru valences are expected to be present on the Au@Ru NPs. DFT calculations were carried out (Supplementary Table 1, Fig. 2d, and Supplementary Fig. 9) to further confirm the attribution of the correlated Raman bands, and it was found that the frequency of the Ru-H stretching mode increases with an increase in the valence state of Ru, which is consistent with previous publication (J. Mol. Struct. 1993, 300, 519-525) that the force constant of Ru-H bond is larger for Ru(n+) with higher electronegativity. The calculated vibrational frequencies of the *H metal-adsorption bonds on Ru(4+), Ru(close to 4+), and Ru(2+) active sites are around 1987 cm^{-1} , 1991 cm^{-1} , and 1947 cm^{-1} , respectively, which match well with our experimental results. This means that the band around 1960 cm^{-1} in our experiment can be assigned to metal-adsorption bond vibration of *H at RuO_x surfaces.

Figure R3 Deuterium isotopic substitution experiment of the HER catalyzed by 55 nm Au@2.5 nm Ru NPs in a 0.1 M NaOD solution (D₂O).

Reviewer #2 (Remarks to the Author):

This manuscript reports on in situ plasmon-enhanced Raman spectroscopy of alkaline HER process at Ru catalyst surfaces. *H and *OH intermediates were directly detected on Ru surfaces and the importance of the valence states of Ru was discussed with the assistance of the DFT simulations. This work provides the microscopic outline of the hydrogen evolution process with strong evidence and thus will be suitable for publication after revisions are noted.

Response: We thank the referee for the very valuable and supportive comments on our manuscript, which help to improve our work. Here are point-by-point answers to his/her comments, with the corresponding changes highlighted in yellow in the revised manuscript and supplementary information (SI).

Comment 1: The Stark tuning rate of OH stretch was smaller on the Ru surface than on Au or Pd surface. This difference was explained by the indirect contact between the Ru surface and water molecules as the presence of the adsorbed *H on the Ru surface. However, the adsorbed *H should similarly exist on the Pd surface in the HER potential region, which seems to be inconsistent with the explanation.

Response 1: We found that the Stark tuning rate of $\nu_{\text{O-H}}$ (3200-3700 cm^{-1}) on the investigated Ru surfaces in the H_2 evolution potential ($-0.95 \text{ V} - +0.05 \text{ V}$) in our work is in the range of 3.6 cm^{-1}/V and 30.2 cm^{-1}/V , which is much smaller than that on the bare Au ($\sim 71 \text{ cm}^{-1}/\text{V}$, Figure R4a, Supplementary Figure 14) (Nat. Mater. 2019, 18, 697-701) and Pd ($\sim 76 \text{ cm}^{-1}/\text{V}$) surfaces (Nature, 2021, 600, 81-85; Chem. Commun. 2007, 44, 4608-4610). Namely, the interfacial water is less sensitive to the electrode potential of the Ru than the Au and Pd, which could be attributed to direct contact between the interfacial water and the Au and Pd surfaces, whereas the indirect contact between the interfacial water and the Ru surfaces owing to the adsorption of *H on the first layer and the interfacial water on the second layer.

Furthermore, we found that the $\nu_{\text{Ru-H}}$ (1800-2000 cm^{-1}) can be observed over the HER potential range (Fig. 2b), whereas the $\nu_{\text{Pd-H}}$ could not be observed (Figure R4b,

Chem. Commun. 2007, 44, 4608-4610; Nature, 2021, 600, 81-85). The probable reason is that H may be rapidly absorbed and embedded into the Pd atomic lattice, resulting in low coverage of H on the Pd surface, which is difficult to be detected (Nat. Mater. 2014, 13, 802-806; J. Am. Chem. Soc. 2014, 136, 10222-10225; J. Phys. Chem. C 2016, 120, 23836-23841;). Furthermore, the bending vibration of water can be observed on Pd surfaces at around 1615 cm^{-1} (Nature, 2021, 600, 81-85; Chem. Commun. 2007, 44, 4608-4610) whereas it is very weak on Ru surfaces. These observations further prove the different behavior of $^*\text{H}$ and interfacial water on Ru and Pd surfaces in the HER potential region.

Figure R4 (a) In situ Raman spectra of the alkaline HER process at 55 nm Au electrode surfaces in a 0.1 M NaOH solution saturated with Ar. (b) The enhanced Raman spectroscopy of surface water on Pt, Pd and Au metals, and propose conceptual models for water adsorbed on Pt Pd and Au metal surfaces. (Chem. Commun. 2007, 44, 4608-4610).

Comment 2: For the peak separation analysis of the OH stretch band, error bars should be indicated because the baseline subtraction may affect the intensities of $\text{Na}^+\text{H}_2\text{O}$ and 4-HB- H_2O .

Response 2: We thank the referee for the kind reminder. The error bars have been added in Figure 4d, Supplementary Figure 12 (c), and Supplementary Figure 13 (c)

Comment 3: The original Ru surfaces are partially oxidized as shown in XPS spectra in Supplementary Fig. 4. This information should be presented before discussing the potential dependence of Raman spectra in Fig. 2. Otherwise, the potential dependence

of the Ru-H band for different Ru valence states is very confusing. The series of Raman spectra shown in Fig. 2 should include not only the potential-induced variations but also the time-course changes. This should be discussed.

Response 3: As the referee suggested, the XPS spectra of the Ru 3p electrons and O 1s electrons of the Au@Ru nanocatalysts have been removed to Supplementary Figure 2, and the related discussion has been removed to “Construction and characterization of Au@Ru core-shell nanoparticles (NPs)” before discussing the potential dependence of Raman spectra in Fig. 2 in “HER process at Ru surfaces under alkaline conditions”. We agree with the referee that the series of Raman spectra shown in Fig. 2 includes not only the potential-induced variations but also the time-course changes (Figure R5). When the control potential is -0.45V , it is found that the content of high-valence Ru-H gradually decreases with the change of time, which is consistent with the gradual reduction of high-valence Ru at the hydrogen evolution potential. It is worth mentioning here that in the initial state, the catalyst surface contains different valence states of Ru. In the HER potential range, the hypervalent Ru is continuously reduced, and its surface behavior is affected by both potential-induced variations and time-course changes. For zero-valent Ru, the surface behavior is mainly affected by potential illustrated by the Stark shift of Ru(0)-H. (J. Electroanal. Chem. 2021, 881, 114955, Ber. Bunsenges. Phys. Chem. 1978, 82, 1046-1050).

Figure R5 In situ Raman spectra of the distinct *H varying with time at -0.45V for 55 nm Au@2.5 nm Ru NPs in a 0.1 M NaOH solution saturated with Ar.

Reviewer #3 (Remarks to the Author):

Comments on the Manuscript: NCOMMS-23-10848-T

The present work presents a deep insight into the HER mechanism on Ru surfaces in alkaline media. By using Au@Ru core-shell nanoparticles, key spectroscopic information was obtained from hydrogen adsorption (H^*), hydroxide adsorption (OH^-), and the water dissociation process. This experimental data combined with theoretical calculation revealed the role of different valence Ru on the surface (+2, +2 - +4, +4) in the adsorption of H, OH, and in the whole HER mechanism, concluding that OH^- adsorbed on RuO_2 serve as an anchoring that helps to transfer protons from the interface to the Ru surfaces, favoring the H^* recombination and leading to higher hydrogen evolution. Findings from Raman signals help to support this observation by funding OH^* signal on RuO_2 surfaces at large negative potentials and two distinct H^* . Overall, the work was well executed, and the experimental design was very well planned with a solid theoretical calculation to support the experimental observations. Despite that, I have some questions and concerns about the work that I would like to clarify. After this, I believe the manuscript needs some minor revisions to address the above comments before it could be considered for publication in Nature Communications.

We thank the referee for agreeing with our views and affirming the importance of the research. His/her expertise and insightful comments are valuable for the improvement of our work. Here are point-by-point answers to his/her comments, with the corresponding changes highlighted in yellow in the revised manuscript.

Comment 1. Why do the authors experiment with this media and not in more tested typical electrolytic conditions such as 1 M KOH? Authors are encouraged to provide a solid reason for this. In any case, have the authors tried performing the experiment in such conditions?

Response 1: We thank the referee for the comments. Electrolytes of NaOH or KOH with concentrations of 0.1 M or 1.0 M are both used for HER investigation, and 1.0 M KOH is a more typical electrolyte. Still, there are many works of literature report on

0.1M electrolytes for alkaline HER mechanism study (Sci. Adv. 2016, 2, e1501602; J. Am. Chem. Soc. 2023, 145, 12051-12058; Angew. Chem. Int. Ed. 2021, 60, 13452-13462; Nano Energy, 2016, 29, 29-36; Adv. Sci. 2018, 5, 1700464; ACS Catal. 2019, 9, 9973-10011; Energy Environ. Mater. 2019, 2, 201–208, etc.). Therefore, we believe that the study under the condition of 0.1 M NaOH is also representative.

We ever performed the experiment in a 1.0 M KOH solution, a more tested typical electrolytic condition, as the referee mentioned. As can be observed in Figure R6, the overpotentials of alkaline HER in 1.0 M KOH and NaOH are similar (67 mV). But they are much smaller than in 0.1 M NaOH (176 mV), indicating the faster reaction kinetics in 1.0 M electrolytes. Accordingly, during the experiment, lots of bubbles were generated in a short time due to the fast HER rate under the 1.0 M KOH condition, especially at extremely negative potentials, which influenced the stability of the system for spectra acquisition. The aggregation state of the Au@Ru NPs even changed during the spectra acquisition sometimes, resulting in the near disappearance of the Raman signal over a potential of -0.45 V, shown in Figure R7. Therefore, we then decided to carry out an experiment in 0.1 M NaOH to guarantee the acquisition of qualified in situ Raman data in the complete HER potential range. It should be noted that the obtained spectra at the initial 5 potentials show similar main important information, like bands and their variations, compared to that obtained in the 0.1 M NaOH. Figure R6 has been added in SI as Supplementary Figure 5. The related explanation has been added and highlighted in “HER process at Ru surfaces under alkaline conditions” in the revised manuscript. To keep consistent, all the electrolytes used in our electrochemical performance and in situ Raman tests are 0.1 M NaOH.

Figure R6 HER polarization curve at 55 nm Au@2.5 nm Ru surfaces in Ar-saturated 0.1 M NaOH (black), 1.0 M KOH (red), and 1.0 M NaOH (blue), 5 mV/s scanning rate and 1,600 r.p.m.

Figure R7 In situ Raman spectra of the alkaline HER process at 55 nm Au@2.5 nm Ru electrode surfaces in an Ar-saturated 1.0 M KOH solution. Spectra were recorded from negative potential to positive potential.

Comment 2: In Figure 2b, as far as I understood, the potential was scanned from more negative to more positive potentials, have the authors tried performing the experiment in the opposite direction? If yes, could the authors comment something on this? Could

be expected differences depending on the direction in which the experiment was performed?

Response 2: Yes, the potential was scanned from more negative to more positive potentials in Figure 2b and we have tried to perform the experiment in both directions (positive to negative and negative to positive). Figure R8 shows HER polarization curves of alkaline HER catalyzed by 55 nm Au@2.5 nm Ru scanned in both directions. The two electrochemical LSV curves show nearly overlapped profiles with tiny differences in overpotential of only around 3 mV at the current density of -10 mA cm^{-2} in a 0.1 M NaOH solution. Namely, the scan direction of the potential has little influence on the electrochemical performance of the system.

We ever tried to perform the in situ Raman experiments from more positive potential to more negative potential shown in Figure R9. We initially observed Raman bands of Ru-O (522 cm^{-1}), Ru-OH (719 cm^{-1}), different Ru-H (1816 cm^{-1} and 1942 cm^{-1}), and interfacial water at -0.05 V , which were consistent with the key species observed when the potential was scanned from more negative potential to more positive potential. However, it was challenging to obtain the qualified spectra when the potential is more negative than -0.65 V due to the disturbance of the generated bubbles. To avoid the disturbance of the bubbles, we started the measurements from the most negative potential and selected the most stable surface region for the data acquisition. Then for the rest of the more positive potentials, the system was stable as well. We took this way to guarantee the acquisition of qualified in situ Raman spectra in the complete HER potential range. Regarding the referee mentioned the differences between the two potential scan ways, as can be observed, although the qualities of the spectra are different, the main important information, like bands and their variations, are similar in the two systems because the HER process is reversible. This has been simply mentioned and highlighted in “HER process at Ru surfaces under alkaline conditions” in the revised manuscript.

Figure R8 HER polarization curve in different scanning directions at 55 nm Au@2.5 nm Ru surfaces in Ar-saturated 0.1 M NaOH.

Figure R9 In situ Raman spectra of the alkaline HER process at 55 nm Au@2.5 nm Ru electrode surfaces in an Ar-saturated 0.1 M NaOH solution. Spectra were recorded from positive potential to negative potential.

Comment 3: Regarding the isotopic labeling experiment, do the authors only perform the experiment at -0.35 V? How about other relevant potentials? For instance, what's the comparison with the behavior of the signals at a more positive potential, i.e., closer to the onset potential? Is there a different behavior? or the comparison at a more negative potential, i.e., -0.8 V. It would be interesting to have this comparison with the isotopic labeling experiment, so a deeper generalization could be extracted.

Response 3: We agree with the referee that the isotopic labeling experiment performed in the complete HER potential range ($-0.95\text{ V} - +0.05\text{ V}$) is helpful for the extraction of a deeper generalization. We have performed in situ D_2O experiments in the complete HER potential range as the H_2O experiment in Figure 2b to provide sufficient data for this purpose, according to the referee's suggestion. As shown in Figure R10 (Supplementary Figure 6), during the HER potential range from -0.95 V to 0.05 V , generally, the changing trend of the Raman spectra (appearing or disappearing of the bands, position, and intensity of the existing bands) in the D_2O experiment is similar to that in H_2O . For example, in the isotope experiment, we also observed the Raman vibration signals of Ru-O (516 cm^{-1}), Ru-OD (672 cm^{-1}), two distinct D (1318 cm^{-1} and 1411 cm^{-1}), and interfacial water D_2O (2503 cm^{-1}) on Ru at -0.95V . The shifts of these vibrational frequencies accord with the expected shifts from the mass conversion of the formula ($\gamma=71.1\%$) (as calculated, see mass formula section of Supplementary Information) and previous reports in the literature (Nat. Catal. 2021, 4, 711-718, J. Am. Chem. Soc. 2020, 142, 8748-8754). With the potential shifts positively, the intensity of $^*\text{D}$ on the high-valence Ru (1411 cm^{-1}) gradually decreases, whereas the frequency of $^*\text{D}$ on zero-valence Ru (1318 cm^{-1}) redshifts. Our isotope experiment further verifies the phenomenon observed in the H_2O experiment. The related explanation has been added and highlighted in "HER process at Ru surfaces under alkaline conditions" in the revised manuscript.

Figure R10 Deuterium isotopic substitution experiment of the HER catalyzed by 55 nm Au@2.5 nm Ru NPs in a 0.1 M NaOD solution (D_2O).

Comment 4: In Supplementary Figure 4, the authors are encouraged to present the O XPS signal as other reports used to present, so a better and more accurate correlation in the types of different valences Ru can be extracted.

Response 4: We thank the referee for the kind suggestion. We have performed the O XPS spectrum (Figure R11) of Au@Ru NPs and presented them in Supplementary Figure 4 (b) according to the referee's suggestion. The O 1s XPS spectrum for Au@Ru NPs can be assigned to lattice oxygen, adsorbed hydroxide, and water ($\text{OH}^-/\text{H}_2\text{O}$) located at 530.4 and 532.5 eV, respectively (Electrochem. Commun. 2007, 9, 239-244, J. Energy Chem. 2021, 54, 510-518), which demonstrates the oxidation of the most out-layer of the Ru shell to $\text{RuO}_2(4+)$ under atmospheric conditions. Therefore, different Ru valences are expected to be present on the Au@Ru NPs. The explanation has been added and highlighted in "Construction and characterization of Au@Ru core-shell nanoparticles (NPs)" in the revised manuscript.

Figure R11 XPS spectra of the O 1s electrons of Au@ Ru nanocatalysts.

Comment 5: Authors claim that "With potential increasing, the vibrational frequency of the $\text{Ru}(n+)$ -H Raman band (red square) around 1960 cm^{-1} red-shifts and its intensity (blue square) gradually decreases, which could be due to the gradual reduction of the high valence $\text{Ru}(n+)$ into zero-valent $\text{Ru}(0)$ under the reduction potential". I have some serious concerns about this expression and interpretation of the data. I would expect this behavior to be more pronounced at more negative potentials, not at more positive ones as the author shows in their results. In general, at more negative potential I expect that Ru is more in the form of $\text{Ru}(0)$ than in the form of a high valence Ru. Could the

authors explain this fact?

Response 5: We thank the referee for his/her expertise and kind suggestion. As shown in Figure R12a, the CV curve of 55 nm Au@2.5 nm Ru NPs shows two pairs of redox peaks at around 0.47/0.63 V and 1.09/1.24 V attributed to the Ru(+3)/Ru(+4) and Ru(+4)/Ru(+6) (Trans. IMF 2007, 85, 194-201, Electrochim. Acta 2020, 354: 136625, Chin. J. Catal. 2022, 43, 130-138). At the hydrogen evolution potential, Ru (+2) is gradually reduced to Ru (0) when the potential is negative than around -0.2 V (Figure R12b) (Energy Environ. Sci. 2021, 14, 5433; J. Phys. Chem. B 2004, 108, 12898-12903; Ber. Bunsenges. Phys. Chem. 1978, 82,1046-1050). The original Ru surface was partially oxidized (Ru(n+)) in our system. When we started the in situ Raman data acquisition from a negative potential to a more positive potential, the Ru surface could not be completely reduced to Ru (0) immediately. Ru(0) and Ru(n+) should coexist, two distinct Ru-H, therefore, were observed. During the spectra acquisition from more negative potential to more positive potential, although the potential increased, they were still at negative HER potentials. The Ru(n+) therefore was still gradually reduced to Ru(0). Higher valence Ru(n+) corresponds to higher wavenumber Ru-H. The reduction of Ru(n+) to Ru(0) results in the red-shift of the vibrational frequency of the Ru-H as we discussed and highlighted in the revised manuscript in “HER process at Ru surfaces under alkaline conditions”.

Figure R12 (a) Cyclic voltammogram curves of 55 nm Au@2.5 nm Ru NPs in a solution of 0.1M NaOH. (b) Proposed Surface Reactions of RuO₂/Ti Electrodes (Ber. Bunsenges. Phys. Chem. 1978, 82,1046-1050).

Comment 6: Regarding Figure 2b and related to the previous comment. It would be interesting to see how the evolution of the band corresponding to Ru-O (516 cm^{-1}) with

the potential, plotted in a similar way that has been done for H signals. The representation of this evolution will give a better understanding of the composition of the different valence Ru on the surface as the potential is being scanned.

Response 6: The normalized Raman intensity and frequency shifts of the Ru-O band versus the HER potentials have been plotted and shown in Figure R13. In the potential range of -0.95 V to -0.35 V, the intensity of the Ru-O band decreases whereas its position keeps constant, which is consistent with the changing of the Ru(n+)-H. This further indicates that the higher wavenumber Ru-H Raman band is related to the higher valence state Ru. As the potential shifts positively, the degree of Ru-O reduction decreases, leading to the increase of the Ru-O content till the ending potential of 0.05 V. Figure R13 has been added in SI as Supplementary Figure 8, and the related explanation has been added and highlighted in “HER process at Ru surfaces under alkaline conditions” in the revised manuscript.

Figure R13 Normalized Raman intensities (blue) and frequency shifts (red) of the Ru-O band at Ru surfaces in the HER potential range. The solution is 0.1 M NaOH saturated with Ar.

Comment 7: In the expression “We performed CVs characterizations on Au@Ru catalysts in a non-faradaic area at different scan rates to determine the ECSA” change the non-faradaic area by “non-faradaic region”.

Response 7: We greatly thank the referee for the kind reminder. We have changed the “non-faradaic area” to “non-faradaic region” and highlighted it in the revised manuscript.

Comment 8: I guess the ECSA study was conducted after the electrochemical experiment so that the same amount of catalyst was drop cast. if it was like this, I encourage the author to mention it in the text so the reader could relate the area with the actual LSV experiment shown for HER with the corresponding ECSA.

Response 8: We greatly thank the referee for the kind suggestion. We have added and highlighted the description “The ECSA study was conducted after the electrochemical experiment so that the same amount of catalyst was drop cast” in “HER performance at Ru surfaces in different valence states under alkaline conditions” in the revised manuscript.

Comment 9: Regarding ECSA calculations, me is confusing that the authors oxidize the electrode at +0.95 V, however, the study of the ECSA was done in a range between +0.80 V and + 1.0 V. Maybe the authors made a mistake in the potential region. If not, the authors should explain why using this potential value.

Response 9: We thank the referee for his/her expertise and kind suggestion. We agree with the referee that our previously selected potential range (+0.80 V – + 1.0 V) was not proper for the ECSA study because of the oxidation reaction occurred here. It is worth mentioning that the selection of a non-Faraday current region for Ru, a metal with multiple valence states, is difficult. Thanks for the referee’s suggestion, we made the CV measurement in the tested electrolyte, 0.1 M NaOH (Figure R14), and found that there is no obvious redox reaction in the potential range of +0.1 – +0.20V (Small 2021, 17, 2101163). The study of the ECSA was then done in this potential range and the results are shown in Figure R15, which has been updated to the SI as Supplementary Figure 10 and the related description has been revised and highlighted in “HER performance at Ru surfaces in different valence states” in the revised manuscript.

Figure R14 Cyclic voltammogram curves of 55 nm Au@2.5 nm Ru NPs in a solution of 0.1M NaOH.

Figure R15 Electrochemical active surface area (ECSA): Cyclic voltammograms of different valence Ru in a non-faradaic region at different scan rates, (a) Original, (b) Reduced, (c) Oxidized. (d) Scan rate dependence of the current at 0.15 V vs. RHE.

REVIEWERS' COMMENTS

Reviewer #1 (Remarks to the Author):

I have now read the rebuttal letter as well as the revised version of the article and based on the comments/revisions made I am pleased to recommend the work for publication in Nature Communications.

Reviewer #2 (Remarks to the Author):

The authors provided convincing responses to the questions raised by this reviewer in the rebuttal, but some information they presented are not properly included in the main text or in Supplementary Information.

(1) Regarding the Stark tuning rate of OH stretch, the absence of the Pd-H stretch due to the H absorption should be explained in the main text; otherwise, it is difficult to understand the similar trend in the Stark tuning rates between Au and Pd. It is also needed to discuss whether the Stark tuning on Ru is similar to that on Pt.

(2) For the peak separation analysis, the authors added the error bars in the revised figures. However, this reviewer cannot understand why the error bars can be so small when the very broad and featureless OH stretching peaks are divided into three components.

(3) Fig. R5 should be presented in S.I. because this information is helpful not only to strength the vibrational assignments of *H species but also to clarify the connection between Fig. 2 and Fig. 3. In this sense, the time-course change should also be presented for the oxidation condition. In the meantime, why is the reduction reaction so slow in the 2-nm thick Ru shells?

Reviewer #3 (Remarks to the Author):

After reading the responses provided by the authors, I believe they have addressed most of the concerns adequately and correctly. I acknowledge that the authors have made efforts to improve the manuscript and clarify some key points to the reader. After these changes, I believe that the manuscript is ready to be considered for publication in Nature Communication. However, I still have a minor concern that I would like to clarify:

1. Regarding Figure 3c, did the authors calculate the current density based on the ECSA values or on the geometric area? I think it is very important to specify this, as the difference between using one or the other could lead to significant changes. Perhaps I missed it in the manuscript, but if it is not specified, I think it is important information to clarify.

Point-by-point response to the Reviewers' comments

Reviewer #1 (Remarks to the Author):

I have now read the rebuttal letter as well as the revised version of the article and based on the comments/revisions made I am pleased to recommend the work for publication in Nature Communications.

Response: We once again express our gratitude to the reviewer for his/her expertise and valuable suggestions on improving our manuscript, as well as his/her final recommendation for the publication of our work in Nature Communications.

Reviewer #2 (Remarks to the Author):

The authors provided convincing responses to the questions raised by this reviewer in the rebuttal, but some information they presented are not properly included in the main text or in Supplementary Information.

Response: We thank the reviewer for confirming our responses to the questions raised by his/her in the first turn of the reviewer's comments and for providing further feedback. And the following are the point-by-point answers to the comments with the corresponding changes highlighted in yellow in the revised manuscript and supplementary information.

Comment 1: Regarding the Stark tuning rate of OH stretch, the absence of the Pd-H stretch due to the H absorption should be explained in the main text; otherwise, it is difficult to understand the similar trend in the Stark tuning rates between Au and Pd. It is also needed to discuss whether the Stark tuning on Ru is similar to that on Pt.

Response 1: We appreciate and concur with the reviewer's comments regarding the absence of the Pd-H stretch due to the H absorption. Additionally, we find that the Stark tuning rate of the OH stretch on Ru is similar to that on Pt. The low sensitivity of the interfacial water to the potential of the Ru and Pt electrodes, along with the small Stark tuning rate of the OH stretch, can be attributed to the indirect contact between them due to the adsorption of *H on the first layer of the Ru/Pt surface, while the interfacial water level is in the second layer. The above corresponding discussions have been added and highlighted in the section of "The behavior of interfacial water and its effect on alkaline HER" in the revised manuscript (Page 14, line 271).

Comment 2: For the peak separation analysis, the authors added the error bars in the revised figures. However, this reviewer cannot understand why the error bars can be so small when the very broad and featureless OH stretching peaks are divided into three components.

Response 2: We thank the reviewer for the comments. For the Raman test, we obtained stable Raman signals and excellent reproducibility by carefully controlling various experimental conditions. First, we synthesized homogeneous and pinhole-free Au@Ru NPs, and then an appropriate amount of Au@Ru NPs was dropped on the electrode surface to form a uniform SERS film substrate, ensuring consistent Raman enhancement ability across different positions. During Raman testing, we controlled the distance between the optical window sheet and the electrode surface, adjusted the laser power to an appropriate level, and carefully determined the spectrum collection time, etc. The above operations allowed us to obtain stable and high-quality Raman spectra. Second, for the analysis of OH stretch peak separation, we determined full width at half maxima (FWHM) for the same type Raman peak of interfacial water to make it more reasonable and representative and the analysis data present the relative intensity after normalization. Therefore, the error bars we added in the revised figures are relatively small based on these factors. In fact, the standard deviation ranged from $\pm 0.2\% \sim \pm 1.2\%$ in our current work, which is comparable to that reported in our previous publication ($\pm 0.5\% \sim \pm 1.5\%$) (Nature, 2021, 600, 81-85).

Comment 3: Fig. R5 should be presented in S.I. because this information is helpful not only to strength the vibrational assignments of *H species but also to clarify the

connection between Fig. 2 and Fig. 3. In this sense, the time-course change should also be presented for the oxidation condition. In the meantime, why is the reduction reaction so slow in the 2-nm thick Ru shells?

Response 3: We thank the reviewer for the comments and suggestions. As the reviewer suggested, Figure R5 has been presented in Figure 8 of the revised supplementary information, and related discussion has been added and highlighted in the “HER process at Ru surfaces under alkaline conditions” section in the revised manuscript (Page 8, line 160). Meanwhile, as shown in Figure R1 below, under the oxidation condition, the Raman peak intensity of RuO₂ at around 315 cm⁻¹, 463 cm⁻¹, and 685 cm⁻¹ increases with time, and the oxidation is basically complete after 20 min. It is noteworthy that we focus on observing the changes in Ru oxide formation under oxidizing conditions, and hence, we only present spectral information in the low-wavenumber range. The time-course change of the RuO₂ Raman peak intensity for Au@Ru NPs under oxidizing condition has been presented in Figure 11 of the revised supplementary information.

Figure R1 The time-course change of the RuO₂ Raman peak intensity of Au@Ru NPs in a 0.1 M NaOH solution at +0.95 V.

Regarding the slow reduction reaction in the 2 nm-thick Ru shell, which may be attributed to the following two reasons: On the one hand, as shown in Figure R2a, the CV curve of 55 nm Au@2.5 nm Ru NPs shows two pairs of redox peaks at around 0.47/0.63 V and 1.09/1.24 V attributed to the Ru(+3)/Ru(+4) and Ru(+4)/Ru(+6) (Trans. IMF 2007, 85, 194-201, Electrochim. Acta 2020, 354: 136625, Chin. J. Catal. 2022, 43, 130-138). At the hydrogen evolution potential, Ru (+2) is gradually reduced to Ru (0) when the potential is negative than around -0.2 V (Figure R2b) (Energy Environ. Sci. 2021, 14, 5433; J. Phys. Chem. B 2004, 108, 12898-12903; Ber. Bunsenges. Phys. Chem. 1978, 82,1046-1050). The original Ru surface was partially oxidized (Ru(n+)) in our current system. When we applied a potential of -0.45 V, the overpotential of Ru (0) and Ru (+2) valence transition was not very large relatively, so the Ru(n+) surface could not be reduced entirely to Ru (0) immediately. On the other hand, under the hydrogen evolution potential, Ru oxides gradually reduced and evolved into RuO(OH)₂ structure, while both amorphous Ru oxides and Ru hydroxide exhibit relatively poor electrical conductivity. Due to the continuous deprotonation of water molecules to produce hydrogen, the local pH on the catalyst's surface increased. The high OH⁻ concentration environment may help to stabilize the RuO(OH)₂ structure,

resulting in a slow reduction of the high-valence Ru state. (J. Phys. Chem. C 2014, 118, 15315-15323; J. Electroanal. Chem., 2021, 881, 114955)

Figure R2 (a) Cyclic voltammogram curves of 55 nm Au@2.5 nm Ru NPs in a solution of 0.1M NaOH. (b) Proposed Surface Reactions of RuO₂/Ti Electrodes (Ber. Bunsenges. Phys. Chem. 1978, 82,1046-1050).

Reviewer #3 (Remarks to the Author):

After reading the responses provided by the authors, I believe they have addressed most of the concerns adequately and correctly. I acknowledge that the authors have made efforts to improve the manuscript and clarify some key points to the reader. After these changes, I believe that the manuscript is ready to be considered for publication in Nature Communication. However, I still have a minor concern that I would like to clarify:

Response: We thank the reviewer for his/her affirmation of our revised manuscript. His/her last minor concern is valuable for improving our work. The following is response to his/her comment and is refined accordingly in the revised manuscript.

Comment 1. Regarding Figure 3c, did the authors calculate the current density based on the ECSA values or on the geometric area? I think it is very important to specify this, as the difference between using one or the other could lead to significant changes. Perhaps I missed it in the manuscript, but if it is not specified, I think it is important information to clarify.

Response 1: Thanks a lot for the referee's kind reminder. In this work, we calculated the current density in Figure 2a, Figure 3c, and Supplementary Figure 5 in terms of the geometric area of the electrode, which has been clarified and highlighted in the section "HER process at Ru surfaces under alkaline conditions" in Page 6, line 107 of the revised manuscript as "Here, in all the polarization curves of the HER process, the current density is calculated based on the geometric area of the electrode."